# Coadaptation fostered by the SLIT2-ROBO1 axis facilitates liver metastasis of pancreatic ductal adenocarcinoma

Qing Li[1,2,8], Xiao-Xin Zhang[1,3,8], Li-Peng Hu[1,8], Bo Ni[4,8], Dong-Xue Li[1], Xu Wang [1], Shu-Heng Jiang[1], Hui Li[1], Min-Wei Yang[2], Yong-Sheng Jiang[2], Chun-Jie Xu[4], Xue-Li Zhang [1], Yan-Li Zhang [1], Pei-Qi Huang[1], Qin Yang[1], Yang Zhou[5], Jian-ren Gu[1], Gary Gui-Shan Xiao[6,7], Yong-wei Sun [2] ✉, Jun Li [1] ✉ & Zhi-Gang Zhang [1] ✉

To explore the mechanism of coadaptation and the potential drivers of pancreatic ductal adenocarcinoma (PDAC) metastasis to the liver, we study key molecules involved in this process and their translational value. Premetastatic niche (PMN) and macrometastatic niche (MMN) formation in a mouse model is observed via CT combined with 3D organ reconstruction bioluminescence imaging, and then we screen slit guidance ligand 2 (SLIT2) and its receptor roundabout guidance receptor 1 (ROBO1) as important factors. After we confirm the expression and distribution of SLIT2 and ROBO1 in samples from PDAC patients and several mouse models, we discover that SLIT2-ROBO1-mediated coadaptation facilitated the implantation and outgrowth of PDAC disseminated tumour cells (DTCs) in the liver. We also demonstrate the dependence receptor (DR) characteristics of ROBO1 in a follow-up mechanistic study. A neutralizing antibody targeting ROBO1 significantly attenuate liver metastasis of PDAC by preventing the coadaptation effect. Thus, we demonstrate that coadaptation is supported by the DR characteristics in the PMN and MMN.

Pancreatic ductal adenocarcinoma (PDAC) is only surgically resectable in less than 20% of cases mainly due to the devastating metastatic nature of PDAC, and the liver is the most preferred site for distant metastasis. Distant metastasis to specific target organs requires the coadaptation of disseminated tumour cells (DTCs) derived from primary tumours and target organ microenvironment resident cells[1,2]. The "seed and soil" theory hypothesizes that the seed (DTCs) and soil (microenvironment of the target organ) both adapt to each other to gain an advantage in achieving metastasis[3]. According to this theory, coadaptation of DTCs and hepatocytes may take place in each step of PDAC liver metastasis to facilitate metastatic niche formation and expansion, but the driving power throughout this process remains to be discovered. To uncover the underlying mechanism, key events in continuous coadaptation involved in liver metastasis, including

[1]State Key Laboratory of Oncogenes and Related Genes, Shanghai Cancer Institute, Ren Ji Hospital, School of Medicine, Shanghai Jiao Tong University, Shanghai 200240, P.R. China. [2]Department of Biliary-Pancreatic Surgery, Ren Ji Hospital, School of Medicine, Shanghai Jiao Tong University, Shanghai, P.R. China. [3]Jiangsu Key Laboratory of Medical Science and Laboratory Medicine, School of Medicine, Jiangsu University, 301 Xuefu Road, Zhenjiang 212013 Jiangsu, P.R. China. [4]Department of Gastrointestinal Surgery, Ren Ji Hospital, School of Medicine, Shanghai Jiao Tong University, No. 1630, Dong Fang Road, Pu Dong New District, Shang Hai 200127 Pu Dong, People's Republic of China. [5]Department of Obstetrics and Gynecology, Shanghai Jiao Tong University Affiliated Sixth People's Hospital, Shanghai 200233, P.R. China. [6]School of Pharmaceutical Science and Technology, Dalian University of Technology, Dalian, P.R. China. [7]Functional Genomics and Proteomics Laboratory, Osteoporosis Research Center, Creighton University Medical Center, Omaha, NE 68131, USA. [8]These authors contributed equally: Qing Li, Xiao-Xin Zhang, Li-Peng Hu, Bo Ni. ✉e-mail: syw0616@126.com; junli@shsci.org; zzhang@shsci.org

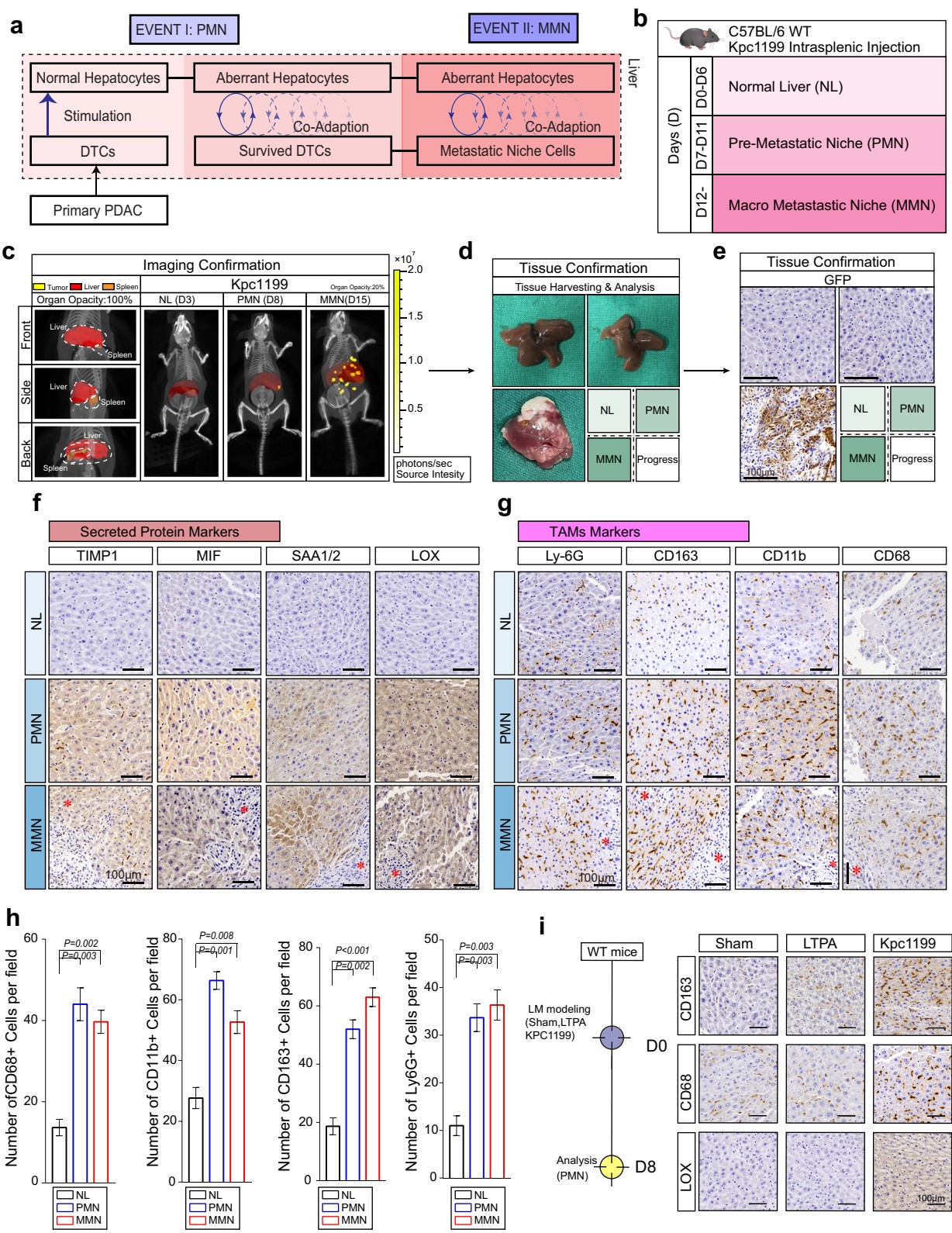

premetastatic niche (PMN) and macrometastatic niche (MMN) formation, were selected for further study (Fig. 1a)[4,5]. Recent studies have revealed that the PMN undergoes conversion to a complex tumour microenvironment (TME) to support the successful colonization of seeds, and the formation of the PMN is recognized as the initial step in metastasis progression[6,7]. For metastasis, native cells in the target organ alter their original secretion protein profiles and begin to recruit TME cells to generate a breeding ground before extravasation and

implantation of tumour cells at the PMN stage, although few morphological changes can be observed at these sites[8,9]. Thus, it is difficult to obtain clinical samples of PMNs formed in the livers of patients, and the means of PMN detection and definition are also limited. Here, we utilized an intrasplenic injection mouse model with relatively permanent PMN and MMN time points in the process of PDAC liver metastasis and further verified our discoveries in the Kras[G12D/+]/Trp53[R172H/+]/Pdx1-Cre (KPC) model and clinical samples.

**Fig. 1 | Definition of coadaptation stages. a** Recognition of the coadaptation process between disseminated tumour cells (DTCs) and hepatocytes in liver metastasis of PDAC. **b** Brief procedure displaying the intrasplenic injection model for the liver metastasis process study. (**c**) Liver metastasis model by intrasplenic injection of murine Kpc1199[luc] cells followed by CT combined with 3D organ reconstruction bioluminescence imaging to assess the time point. Mice that received an intrasplenic injection were photographed to detect the normal liver (NL), (Day 3, D3), premetastatic niche (PMN), (Day 8, D8) and macrometastatic niche (MMN) (Day 15, MMN). Red, reconstructed liver; orange, reconstructed spleen; yellow, signals of tumour niches. Reconstructed organ opacity, left, 100%; right, 20%. Scale colour bar: $2.00 \times 10^5 - 2.00 \times 10^7$ ($n = 5$ mice per group). **d** Representative livers from liver metastasis mouse models showing an NL (D3), PMN (D8) or MMN (D15) after injection ($n = 5$ mice per group). **e** IHC-P staining of GFP in left lobules from mouse models showing an NL (D3), PMN (D8) or MMN (D15) ($n = 5$ samples per group, 3 fields assessed per sample). Scale bars, 100 μm.

**f** Representative IHC-P staining of the PMN-specific secreted protein markers LOX, SAA1/2, MIF or TIMP1 in the left lobules of livers from model mice to detect an NL (D3), PMN (D8) or MMN (D15) ($n = 5$ samples per group, 3 fields assessed per sample). Scale bars, 100 μm; * represents metastatic niches. **g, h** Representative IHC-P staining of PMNs expressing the tumour-associated macrophage (TAM) markers CD68, CD11b, CD163 and Ly-6G in the left lobules of livers from mouse models to detect an NL (D3), PMN (D8) or MMN (D15) ($n = 5$ samples per group, 3 fields assessed per sample). Scale bars, 100 μm; * represents metastatic niches. ($n = 5$ samples per group, mean ± SEM, two-tailed unpaired $t$ test, 3 fields assessed per sample). **i** Representative IHC-P staining of the PMN-specific secreted protein marker LOX and TAM markers CD68 and CD163 in the left lobules of livers from model mice on D8 bearing sham, LTPA modelling and Kpc1199 modelling ($n = 5$ samples per group, 3 fields assessed per sample). Scale bars, 100 μm. Source data are provided in the Source Data file.

Members of the axon guidance (AG) family have been considered to play vital roles in a variety of physiological processes, such as embryonic development and neuron growth[10,11]. Accumulating evidence has also revealed their indispensable functions in cancer[12,13]. Intriguingly, several important receptors of the AG family were reported to share common characteristics in different cancer types and were termed dependence receptors (DRs)[14–16]. DRs are special receptors that act as a two-way switch to trigger cell proliferation and survival in the presence of their ligands or induce apoptosis once ligands are removed rather than simply turning on and off. Under normal conditions, it is generally supposed that DRs are responsible for maintaining tissue homeostasis and preventing the escape of cells from their native location[15,17]. For example, one of the AG receptor members, Plexin D1, has been demonstrated to regulate tumour cell survival through DR function, and its ligand Semaphorin 3E could inhibit its apoptotic pathway to facilitate the proliferation of tumour cells;[14] netrin could hamper tumour cell apoptosis by binding to its receptor UNC5;[16] and EphA4 mediated apoptosis in the absence of its ligand EphrinB3[18]. There are also non-AG members that play roles in DR. Cell-adhesion molecule-related/downregulated by oncogenes (CDON) was reported as a receptor of Sonic Hedgehog (SHH); it could recruit and thus activate caspase-9 to trigger apoptosis of colorectal cancer cells, and this effect would be abolished in the presence of SHH[19]. Therefore, it is of interest to explore the mechanisms involved in DR-mediated metastasis. Although discussed in many previous studies, the role of DRs in tumorigenesis, especially in metastasis, remains largely unknown.

In this study, we use our models to filter out the AG molecule SLIT2, which is overexpressed by hepatocytes through progression of the PMN to the MMN, and demonstrate that its classic receptor ROBO1 is a DR. We also find that hepatocyte-derived SLIT2 forms a supportive environment for the implantation of ROBO1-positive DTCs, thus triggering coadaptation of hepatocytes and tumour cells to further support the development of the MMN. Interference with the interaction of these two molecules by neutralizing antibodies significantly hampers the progression of liver metastasis. Our results reveal a role of DRs in cancer metastasis and suggest that SLIT2-ROBO1 axis-mediated coadaptation of "seed" and "soil" is critical in implantation and outgrowth in the PMN until the MMN is formed. We also provide a potential therapeutic strategy to inhibit coadaptation by targeting DRs to not only prevent the proliferation of tumour cells but also trigger their apoptosis.

## Results

### Detection and verification of the PMN and MMN in mouse models

First, we utilized an intrasplenic injection mouse model to study alterations in the liver microenvironment from the PMN state to the MMN state. In this model, injected tumour cells could generate metastatic liver lesions by passing through the portal vein to simulate

PDAC liver metastasis, which is highly reproducible and makes it possible for us to analyse the step-by-step changes in expression profiles in the liver[20,21].

The KPC mouse-derived PDAC cell line Kpc1199 and murine PDAC cell line Panc02 were used to generate liver metastases. Thus, we could cite the approximate time of the metastasis process for Kpc1199 cells (D7-D11 for PMN and D12 + for MMN) and Panc02 cells (D5-D8 for PMN and D9 + for MMN) in the intrasplenic injection model (Fig. 1b, Supplementary Fig. 1a; detailed methods for determination of the PMN and MMN time points are shown in *Materials and Methods*). CT combined with 3D organ reconstruction bioluminescence imaging provided the possibility of observation and precise localization of tiny signal points of metastatic niches (Fig. 1c–e; Supplementary Fig. 1a, b). At D8 for Kpc1199 cells or D6 for Panc02 cells, the PMN had formed, as indicating by a low number of GFP+ cells and expression of the reported markers LOX, SAA1/2, TIMP-1 and MIF; macrophage markers CD68, CD11b and CD163; and neutrophil marker Ly-6G. At D15 for Kpc1199 cells and D12 for Panc02 cells, IHC-detectable metastatic clones were observed in the liver, so this time point was recognized as the metastasis time point (Figs. 1f–h, 2f; Supplementary Figs. 1c–k, 3a)[8,22–27]. We selected D3 instead of D0 as the control group to eliminate the interference of nontumour inflammation due to surgery among the groups. Additionally, we used the sham group and LTPA, a murine PDAC-derived cell line that could not form liver metastases through intrasplenic injection, to confirm that PMN and MMN markers would not exist in the absence of DTCs (Fig. 1i). The time point and location of metastases formed by PDAC cells in the left hepatic lobule were relatively reproducible due to the haemodynamic characteristics of portal veins. In summary, we could thus obtain liver tissues undergoing PMN and MMN formation in mouse models for further exploration.

### Enrichment of hepatocyte-derived SLIT2 in ROBO1-positive tumour cells in liver metastasis progression

Hence, to comprehensively investigate the differentially expressed genes in hepatocytes during the process of PDAC metastasis, we prepared $2 \times 2 \times 2$ mm³ slices of liver tissue without tumours and close to portal veins in the left hepatic lobule of mice at D3, D8 or D15 for Kpc1199 cells and D3, D6 or D12 for Panc02 cells after intrasplenic injection. These specimens were further confirmed by positive staining of PMN markers before being used for further studies (Fig. 2a, b). Then, we performed transcriptomic analysis of these three groups. Gene set enrichment analysis (GSEA) of transcriptome data comparing NL vs. PMN or NL vs. MMN highlighted the importance of the axon guidance pathway in liver metastasis (Fig. 2c). Further assessment of secreted proteins in the axon guidance family revealed common genes that were upregulated in both the PMN- and MMN-detected livers (*Sema3e*, *Slit2* and *Efna4*) (Supplementary Fig. 2a–c). Among these axon guidance-secreted ligands, SLIT2 was filtered out in both the PMN and MMN in the two cell line-derived mouse models (Fig. 2d;

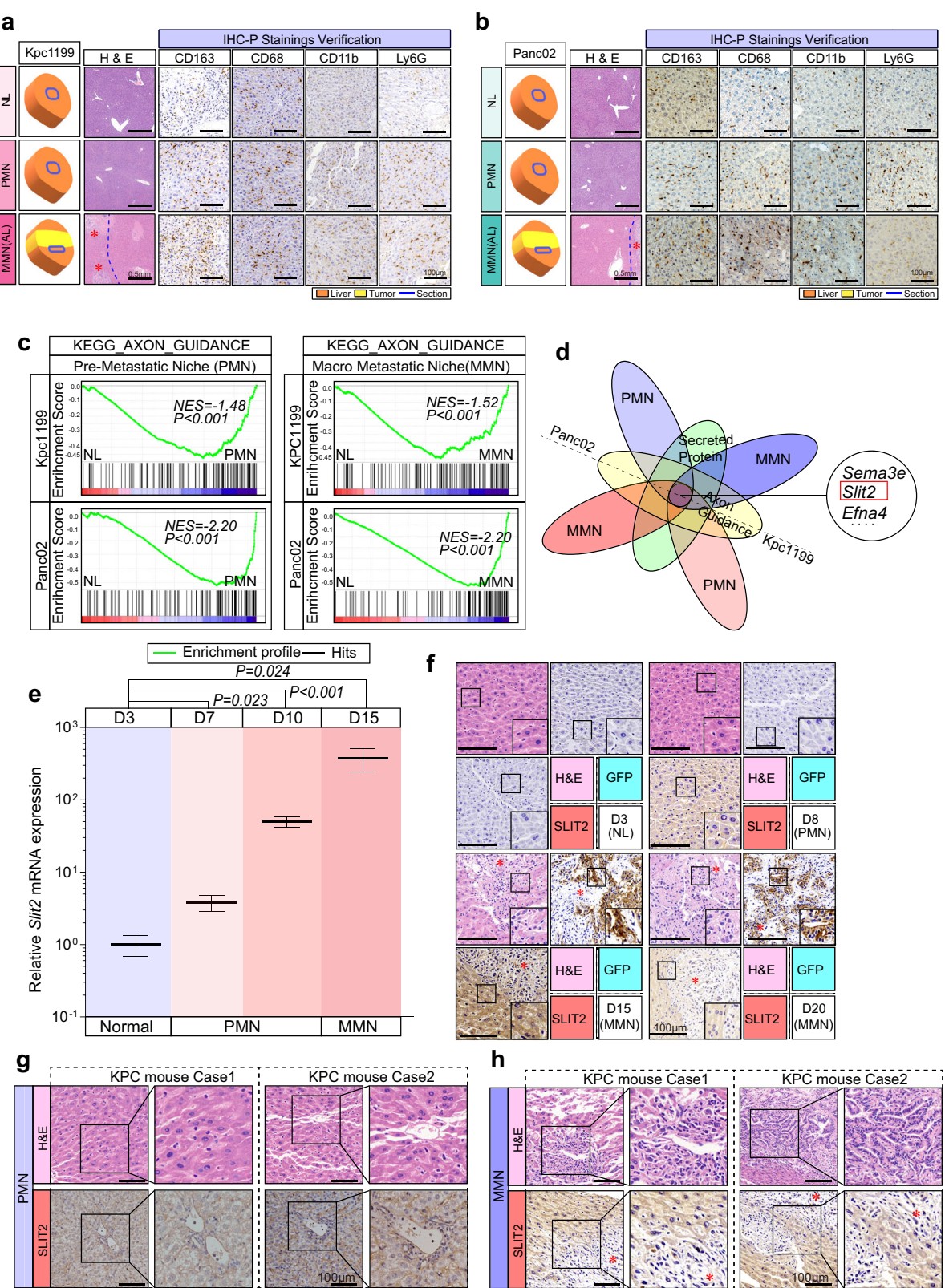

Supplementary Fig. 2a–c). Further results showed that SLIT2 expression began to arise during PMN formation (D3-D7) and lasted until metastatic colonization was complete (D10-D15) in the adjacent area of the metastatic niche (Fig. 2e, f). The same phenomenon was observed in the Panc02-derived liver metastasis model (Supplementary Fig. 3a). To further confirm this, we used KPC mice, which can spontaneously generate PDAC liver metastases. We first detected DTCs in the left

hepatic lobule by measuring the PDAC DTC markers CK19, CD133 and CD44 in KPC mice before 12 weeks of age, at which point metastasis frequently occurred[28,29] (Supplementary Fig. 3b). The PMN markers mentioned above in the intrasplenic injection model were also detected in the livers of KPC mice with DTCs (Supplementary Fig. 3c). Furthermore, high SLIT2 expression was found in DTC-detected PMN- or MMN-formed livers in KPC mice (Fig. 2g, h). To further confirm the

**Fig. 2 | SLIT2 is critical for coadaptation. a**, **b** Livers from intrasplenic injection mouse models induced with Kpc1199 (**a**) or Panc02 (**b**) cells were subjected to transcriptional analysis after confirmation of premetastatic niche (PMN) markers (e.g., CD163, etc.) (*n* = 3 mice per group, 5 fields assessed per sample). Scale bars, 500 μm for H&E staining or 100 μm for IHC-P; * represents metastatic niches. **c** Gene set enrichment analysis (GSEA) based on the gene expression profiles of liver tissues undergoing different metastatic processes after mouse modelling with Panc02 or Kpc1199 cells (false discovery rate analysis; NES, normalized enrichment score). **d** Venn diagram displaying secreted axon guidance genes upregulated in both PMNs and macrometastatic niche (MMN) compared to levels in normal liver (NL) Panc02 and Kpc1199 cells based on the gene expression profiles of liver tissues

from mouse models. **e** Real-time PCR showing the relative mRNA level of SLIT2 in the left lobule of livers from Kpc1199 cell-injected mouse models on D3, D7, D10 or D15 (*n* = 5 per group, mean ± SEM.; two-tailed unpaired *t* test). (**f**) Representative IHC-P staining of H&E, GFP or SLIT2 in the left lobule of livers from Kpc1199 cell-injected mouse models on D3 (NL), D8 (PMN), D15 or D20 (MMN) (*n* = 5 samples per group, 3 fields assessed per sample); * represents metastatic niches. Scale bars, 100 μm. **g**, **h** Representative H&E or IHC-P staining of SLIT2 in PMNs (**g**) or MMNs (**h**) formed in the left lobule of livers from *Kras*^G12D/+/*Trp53*^R172H/+/*Pdx1*-Cre (KPC) mice bearing spontaneous PDAC liver metastasis (*n* = 6 samples per group, 3 fields assessed per sample). Scale bars, 100 μm. Source data are provided in the Source Data file.

source of SLIT2, staining of pathological slices of livers with metastasis from PDAC patients was performed. Adjacent liver of the metastatic niches in PDAC patients exhibited enhanced SLIT2 staining, indicating that SLIT2 was derived from local hepatocytes but not tumour cells (Fig. 3a, b), which was further confirmed by costaining of SLIT2 and albumin as well as *SLIT2* and *ALB* mRNA via in situ RNAscope in both mouse- and human-derived tissues (Fig. 3c, d, f, g).

In our study, we found that aberrantly high SLIT2 expression could participate in the coadaptation of hepatocyte-derived tumour cells and microenvironment cells in liver metastasis progression. Members of the roundabout guidance (ROBO) family, except ROBO4, are the predominant receptors of SLIT2[30,31]. We further discovered that ROBO1, but not ROBO2 or ROBO3, was mainly expressed in metastatic CK19⁺ PDAC cells (Fig. 3e, Supplementary Fig. 3d). Interestingly, we discovered that many more ROBO1-positive metastatic niches could be detected in SLIT2-enriched livers with metastasis in KPC mice, and a strong correlation existed between these two molecules in livers bearing metastases (Fig. 3h, Supplementary Fig. 3e). Moreover, we analysed the expressions of ROBO1 and SLIT2 in GEO datasets GSE71729, and the results revealed a strong correlation between these two molecules in liver metastasis, which further emphasized the vital role of SLIT2-ROBO1 in liver metastasis progression (Supplementary Fig. 3f).

To further assess SLIT2 and ROBO1, we first investigated their expression and roles in primary PDAC. The results demonstrated that SLIT2 is derived from cancer-associated fibroblasts (CAFs) in patients and KPC mouse models but not from ROBO1⁺ tumour cells (Supplementary Fig. 4a, b). Furthermore, the activation of the SLIT2-ROBO1 axis promoted E-cadherin expression and inhibited N-cadherin expression and CDC42 activation to facilitate tumour cell implantation but not migration, in accordance with a previous study (Supplementary Fig. 4c)[32]. Further pathway analysis of GEO Datasets GSE15471 and TCGA PAAD database according to ROBO1 expression further confirmed our findings (Supplementary Fig. 4d, e). To comprehensively analyse the relationship between prognosis and the expression of these two molecules, we examined the functions of SLIT2-ROBO1 in primary PDAC by evaluating the prognosis of 266 patients. Interestingly, we discovered that a high level of ROBO1 in SLIT2-rich TMEs leads to a poor prognosis, while the results were the opposite if the TME was deficient in SLIT2 (Supplementary Fig. 4f–h). These prognostic data revealed that further study of this axis would be of value.

Taken together, the above results reveal that hepatocytes can upregulate SLIT2 secretion for the recruitment and aggregation of ROBO1-positive tumour cells, which could be responsible for their coadaptation.

## SLIT2-ROBO1-mediated coadaptation facilitates outgrowth of liver metastatic niches in vivo

We then characterized the ROBO1 molecule. To study the role of the SLIT2-ROBO1 interaction in the coadaptation of cells in PDAC liver metastasis, tumour cells expressing full-length ROBO1 protein (Kpc1199^Robo1-FL or Panc02^Robo1-FL) were prepared (Supplementary Fig. 5a, b). The results demonstrated that ROBO1-FL-expressing tumour cells showed increased proliferation and colony formation ability in the

presence of recombinant SLIT2 (rSLIT2), and this advantage could be abolished by sROBO, a soluble peptide containing the first 2 Ig domains as a ligand-binding trap (Supplementary Fig. 5c–h). Knockdown of ROBO1 in the human cell line PANC1 significantly decreased the tumour burden in a liver metastasis mouse model (Supplementary Fig. 5i–k).

We next generated *Slit2* hepatocyte-specific conditional knockout mice (*Slit2*^fl/fl/*Alb*-Cre, CKO). To rescue, lentivirus carrying loxp-*Slit2* was used to specifically express SLIT2 in hepatocytes and was administered 14 days before modelling (CKO-RE). We also used lentivirus carrying loxp-*Slit2-GFP*, which could express the SLIT2-GFP fusion protein, to confirm that SLIT2 could be expressed until the end of the experiment (Supplementary Fig. 6c). Further validation of this system was performed through WB and in situ RNAscope (Supplementary Fig. 6a, d, b). Tumour cells expressing full-length ROBO1 protein (Kpc1199^Robo1-FL or Panc02^Robo1-FL) were intrasplenically implanted into *Slit2*^fl/fl (CTRL), CKO and CKO-RE mice (Fig. 4a; Supplementary Figs. 6b, 7a). The results illustrated far fewer metastatic colonies in the CKO group than in the CTRL group. In the CKO-RE group, the paracrine resumption of SLIT2 expression by hepatocytes restored the outgrowth ability of *Robo1*-FL-expressing tumour cells (Fig. 4b, c; Supplementary Figs. 6e–h, 7c–f). Furthermore, tissues from CKO mice exhibited smaller metastatic niches and less Ki67 staining than tissues from CTRL or CKO-RE mice (Supplementary Figs. 6h–k, 7f–i). Interestingly, tumour cells expressing *Robo1*-FL exhibited apoptosis when SLIT2 was lacking in the microenvironment.

To inhibit the direct interaction of SLIT2 and ROBO1, we produced a ROBO1 neutralizing antibody specifically targeting the Ig1-Ig2 domain (Supplementary Fig. 5a). Validation of this neutralizing antibody showed that administration of 500 μg per mouse significantly hampered Kpc1199^Robo1-FL cell-formed liver metastatic niches in a mouse model (Supplementary Fig. 8a, b). Administration of neutralizing antibody every 3 days increased the median survival time of the mouse model to 36 days compared to every 5 days or only once (Supplementary Fig. 8c). The results revealed that administration of the neutralizing antibody sufficiently retarded the progression of metastasis mediated by *Robo1*-FL-expressing tumour cells (Fig. 4e, Supplementary Fig. 9a–e). Further analyses demonstrated that antibody administration significantly reduced the proliferation and increased the apoptosis of metastasized PDAC cells (Supplementary Fig. 9f, g). We have also consolidated these results in PANC1 model mice (Supplementary Fig. 8d–f).

To obtain direct evidence that SLIT2-ROBO1 mediates the coadaptation process, we performed antibody treatment in KPC mice. In this model, a high frequency of spontaneous liver MMN occurrence was observed at approximately 15-16 weeks of age in mice. To prohibit the formation of PMNs in the liver, we started antibody administration at week 12 (Fig. 4f). The data demonstrated that antibody treatment significantly hampered the outgrowth of metastatic niches in the left lobules and decreased the percentage of ROBO1⁺ PDAC cells in these niches (Fig. 4g–i). Interestingly, the accumulation of SLIT2 in adjacent hepatocytes was significantly attenuated, indicating that disturbing ROBO1 function in this axis could significantly affect SLIT2 expression. Furthermore, it was more effective to perform treatment at the PMN stage than at the MMN stage because early administration better

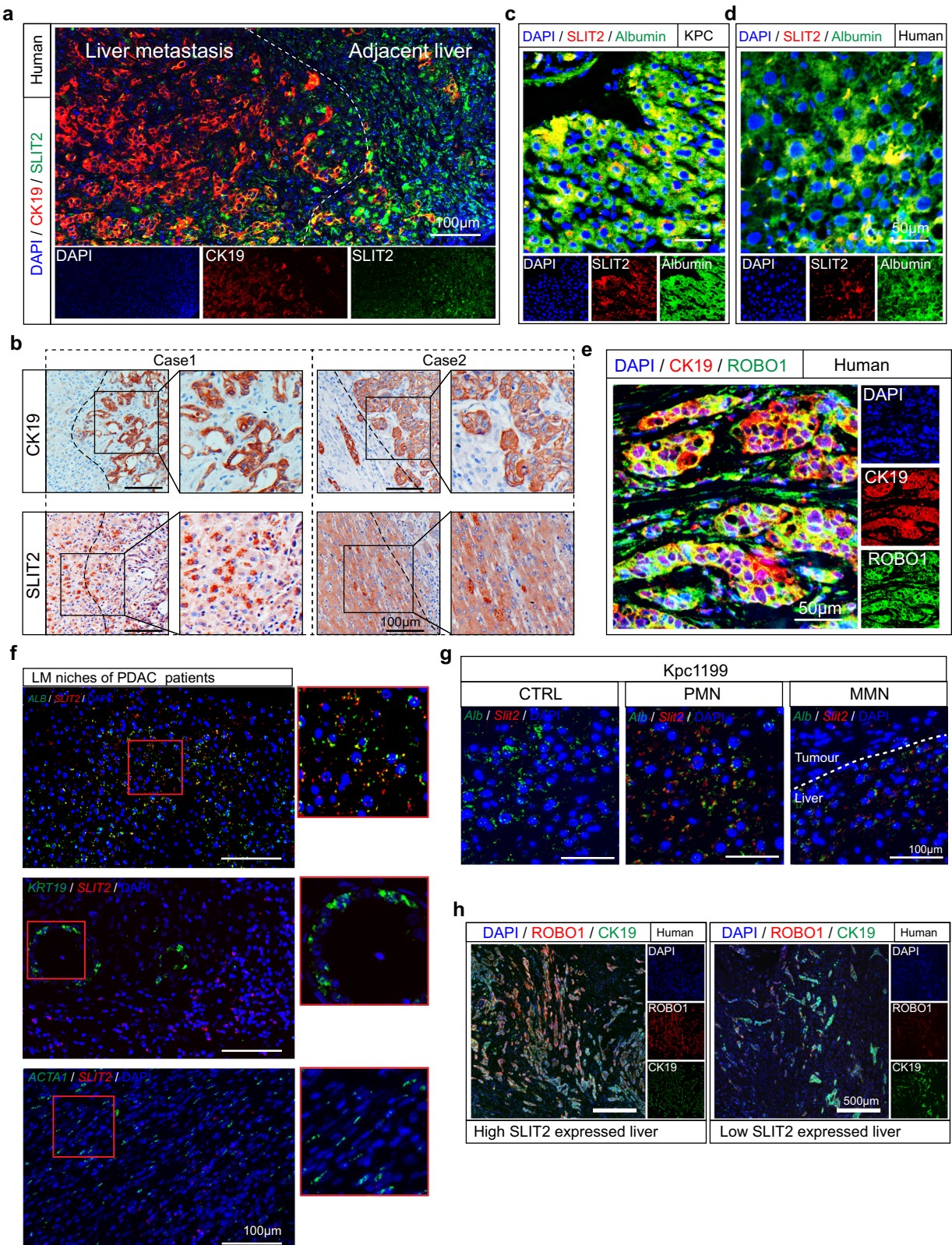

prevented micro niche (≤500 μm) formation in the liver (Fig. 4k, l). Moreover, interference of the coadaptation loop was more efficient when antibody was administered at the PMN stage to inhibit SLIT2 expression (Fig. 4m).

These results highlight the significance of the SLIT2-ROBO1 axis in liver metastasis, while the role it plays in coadaptation needs to be further investigated.

## The SLIT2-ROBO1 axis is critical for the coadaptation of hepatocytes and tumour cells in liver metastasis progression

We then performed IHC-P staining of ROBO1 in 35 PDAC liver metastasis tissues and their matched primary tumours. The results illustrated high ROBO1 expression in most liver metastasis tissues, even if the positive rate was much lower in matched primary tumours (Fig. 5a, b). Further results in KPC mice also confirmed this conclusion

**Fig. 3 | Detection of SLIT2 and ROBO1 distribution in liver metastasis and its relation to prognosis. a** Representative IF staining showing the location of SLIT2 and CK19 in adjacent liver or liver metastases of PDAC patients. SLIT2, green; CK19, red; DAPI, blue (*n* = 35 patients, 3 fields assessed per sample). Scale bar, 100 μm. **b** Representative IHC-P staining of CK19 and SLIT2 in liver metastatic niches in PDAC patients (*n* = 35 cases, 3 fields assessed per sample). Scale bars, 100 μm. **c, d** Representative IF staining showing the location of SLIT2 and albumin in the PMN in the *Kras*^G12D/+^/*Trp53*^R172H/+^/*Pdx1*-Cre (KPC) mouse model (**c**) or adjacent liver of metastatic niches in PDAC patients (**d**). Albumin, green; SLIT2, red; DAPI, blue (*n* = 35 samples for patients, *n* = 6 samples per group for mouse models, 3 fields assessed per sample). Scale bars, 50 μm. **e** Representative IF staining showing the location of ROBO1 and CK19 in the liver metastasis of PDAC patients. ROBO1, green;

CK19, red; DAPI, blue (*n* = 35 samples, 3 fields assessed per sample). Scale bar, 50 μm. **f** Representative RNAscope staining displaying the location of *SLIT2* mRNA and *ALB* (upper), *KRT19* (middle) and *ACTA1* (bottom) mRNAs in the liver metastasis (LM) of PDAC patients. *ALB*, *KRT19* or *ACTA1*, green; *SLIT2*, red; DAPI, blue (*n* = 5 samples, 3 fields assessed per sample). Scale bar, 100 μm. **g** Representative RNAscope staining displaying the location of *Slit2* mRNA and *Alb* mRNA in the premetastatic niche (PMN) and macrometastatic niche (MMN) of Kpc1199 model mice. *Alb*, green; *Slit2*, red; DAPI, blue (*n* = 5 samples, 3 fields assessed per sample). Scale bar, 100 μm. **h** Representative IF staining displaying the expression of ROBO1 in metastatic livers with or without SLIT2 enrichment. CK19, green; ROBO1, red; DAPI, blue (*n* = 6 mice per group, 3 fields assessed per sample). Scale bar, 500 μm. Source data are provided in the Source Data file.

(Supplementary Fig. 10a). To explore why ROBO1 is enriched in liver metastases and how SLIT2-ROBO1 manipulates the selection of tumour cells in coadaptation, we next used a cell mixture composed of 50% Panc02^Robo1-FL^ and 50% Panc02^CTRL^ cells (PG0) in an intrasplenic injection mouse model. Tumour cells derived from separated liver metastases formed by PG0 cells were then cultured (PG1) before the next model, and PG2 cells were obtained from separated liver metastases formed by PG1 cells (Fig. 5c). Flow cytometry analysis demonstrated an increase in Panc02^Robo1-FL^ cells in the population from PG0 to PG2 in the tumour cell mixture, indicating that ROBO1 facilitated the survival and outgrowth ability of tumour cells for selection in the liver (Fig. 5d, e, Supplementary Fig. 5h). The same results were also obtained using PANC1^shCTRL^ and PANC1^shROBO1^ cells (Supplementary Fig. 10b, c). To explore whether SLIT2 was the driving force, we analysed the protein levels of ROBO1 and SLIT2 via IHC-P in the PMN and MMN in mouse models that received intrasplenic injections of PG0 and PG1 cells. The results revealed that ROBO1 staining in the MMN was much stronger in the PG1 group than in the PG0 group, while SLIT2 staining could be detected in all groups, displaying an increasing pattern from the PMN of PG0 cells to the MMN of PG1 cells (Fig. 5f; Supplementary Fig. 10d). WB analysis of PG1 and PG2 cells also provided similar evidence (Supplementary Fig. 10e). In vitro experiments on PG0 cells also revealed that either Kpc1199^Robo1-FL^ or Panc02^Robo1-FL^ could obtain a proliferation advantage in the presence of SLIT2 (Supplementary Fig. 10f). This coadaptation effect could also be disturbed by antibody treatment (Fig. 5g). Blocking the SLIT2 and ROBO1 interaction significantly decreased the staining rate of both molecules (Fig. 5h). These data indicate that the DTC-induced SLIT2-rich microenvironment exerted selective pressure on DTCs themselves and provided ROBO1⁺ cells with a growth advantage in the liver, thus fulfilling the coadaptation process. To confirm this hypothesis, another tumour cell mixture, Kpc1199^Mix-I^, containing equal amounts of Kpc1199^Robo1-FL/GFP^ and Kpc1199^ΔRobo1/mCherry^ cells, or the mixture Kpc1199^Mix-II^, containing equal amounts of Kpc1199^Robo1-FL/GFP^ and Kpc1199^CTRL/mCherry^ cells, was utilized in *Slit2*/CKO or WT mouse models (Fig. 5i). In WT mice, ROBO1-FL-expressing tumour cells displayed a predominant population in liver metastatic niches, especially in Kpc1199^Mix-I^-modelled cells, during coadaptation with hepatocytes, while in *Slit2*/CKO mice, Kpc1199^CTRL^ cells outcompeted Kpc1199^Robo1-FL^ cells in the absence of SLIT2 (Fig. 5j, k). Moreover, the outgrowth of both Kpc1199^Robo1-FL^ cells and Kpc1199^ΔRobo1^ cells was significantly hampered in the SLIT2-deficient TME. The data obtained from flow cytometry analysis were also consistent with these results (Fig. 5l; Supplementary Fig. 10g).

These phenomena indicate that ROBO-FL-expressing tumour cells and SLIT2-expressing hepatocytes can coadapt and that loss of SLIT2 in the TME or blockade of ROBO1 can not only lead to failure of ROBO1⁺ tumour cells in the context of cell competition but can also lead to their elimination in the liver.

## ROBO1 acts as a DR to exert dual effects on tumour cells
Previous studies have demonstrated that several receptors of the axon guidance family, including DCC, Plexin-D1, UNC5 and EphA4, are

DRs[14,16,18,33]. Our clinical prognosis data also provided evidence to support the idea that ROBO1 might have dual functions in PDAC progression (Supplementary Fig. 4f–h).

To confirm this, we constructed *Robo1* lacking the first two Ig domains (*ΔRobo1*), which are required for the SLIT2 and ROBO1 interaction (Supplementary Fig. 4c, d)[34,35]. Then, tumour cells expressing *Robo1*-FL or *ΔRobo1* were intrasplenically injected into mice to further investigate the role of ROBO1 in liver metastasis (Fig. 6a; Supplementary Figs. 11a, 12a). The results showed that mice injected with Kpc1199^Robo1-FL^ or Panc02^Robo1-FL^ cells exhibited more severe liver metastasis than mice in the Kpc1199^ΔRobo1^ group, Panc02^ΔRobo1^ group or CTRL group (Fig. 6b, c, Supplementary Figs. 11b–d, 12b–d). Interestingly, *ΔRobo1*-expressing tumour cells led to much less metastasis than the other two groups and induced the most apoptosis in the metastatic area among the three groups (Supplementary Figs. 11e–h, 12e–h). The results of survival analysis of these mice were in accordance with the results we obtained previously (Fig. 6d, e).

Based on the above results, we postulated that ROBO1 might be a DR. Examinations of the SLIT2 and ROBO1 expression levels in 10 pancreatic cancer cell lines revealed that SLIT2 seldom displays high expression in PDAC cell lines, especially in liver metastasis-derived CAPAN-1 and cFPAC-1 cells with relatively high ROBO1 expression (Supplementary Fig. 13a, b). SW-1990 cells with low expression of both ROBO1 and SLIT2 were selected for transfection with *ΔROBO1*, *ROBO1*-FL or empty vector for further study. The results revealed that the proliferation capability of SW-1990^ROBO1-FL^ cells was much higher than that of SW-1990^ΔROBO1^ cells when additional recombinant SLIT2 (rSLIT2) was present. Notably, in the absence of rSLIT2, both groups presented slower growth rates than SW-1990^CTRL^ cells (Fig. 6f). Similar results were obtained in Panc02 cells (Supplementary Fig. 5c). Moreover, the promoting effects of rSLIT2 on PDAC cell growth could be abolished by either a neutralizing antibody targeting rSLIT2 or sROBO in both SW-1990 and Panc02 cells (Fig. 6g; Supplementary Figs. 5a, d, 13c, d). Colony formation assays of human SW-1990 cells and murine Panc02 or Kpc1199 cells led to the same results (Fig. 6h, Supplementary Figs. 5e–h, 14a). On the other hand, knockdown of ROBO1 significantly abrogated the growth advantage of PANC-1 and BxPC-3 cells with high expression of both ROBO1 and SLIT2 (Supplementary Fig. 14b). Cell lines that seldom expressed ROBO1 or SLIT2 displayed insensitivity to sh*ROBO1* administration in cell growth (Supplementary Fig. 14c, d). Furthermore, the tumour burden of PANC-1 cells in the subcutaneous xenograft model was significantly reduced after stable knockdown of ROBO1 (Fig. 6i, Supplementary Fig. 14e, f) as well as in the liver metastasis mouse model mentioned before (Supplementary Fig. 5i–k). Consistently, overexpression of SLIT2 in CAPAN-1 cells derived from human PDAC liver metastases with a high level of ROBO1 expression increased the tumour burden (Fig. 6j, Supplementary Fig. 14g–i).

We also confirmed cell apoptosis induced by ROBO1 without SLIT2 binding. Flow cytometry analysis showed that both ΔROBO1 and ROBO1-FL significantly induced cell apoptosis in SW-1990 cells in the absence of SLIT2, while the apoptosis of SW-1990^ROBO1-FL^ cells was reversed by rSLIT2 at a concentration of 10 nM or 30 nM, but the

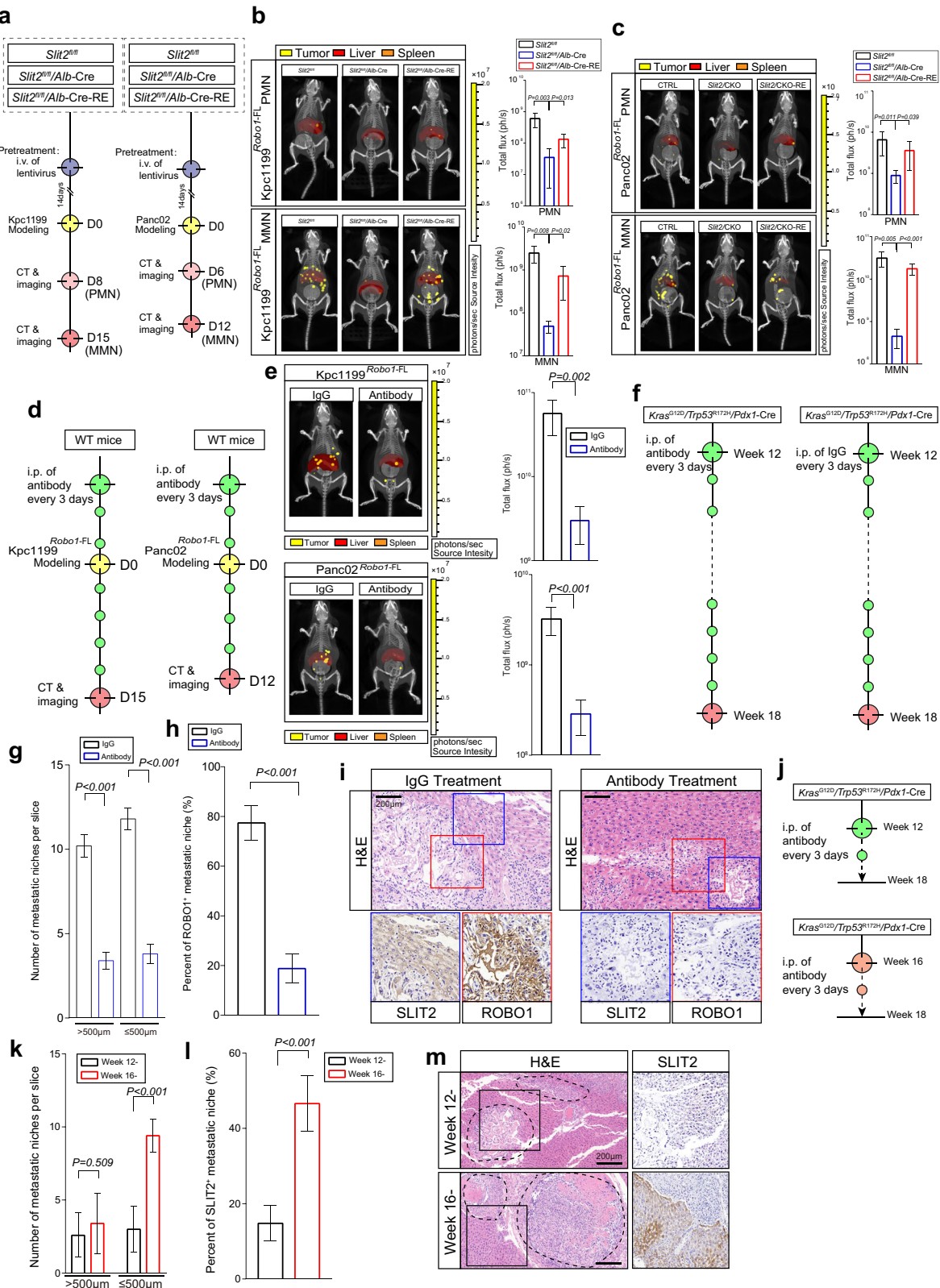

apoptosis of SW-1990$^{\Delta ROBO1}$ cells could not be reversed (Fig. 6k, Supplementary Fig. 15a). Similar results were obtained in TUNEL assays. Moreover, sROBO abolished the anti-apoptotic function of rSLIT2 in SW-1990 cells expressing ROBO1-FL (Fig. 6l, Supplementary Fig. 15b, c). Further anti-cleaved caspase 3 immunofluorescence, a caspase 3/7 kit, and western blotting assays revealed that ΔROBO1 or ROBO1-FL overexpression led to cell death by triggering caspase-

dependent apoptosis via caspase-3, caspase-7, and caspase-9 but not caspase-8 in the absence of SLIT2 (Fig. 6m, Supplementary Fig. 15d). Further in vivo experiments illustrated that in the absence of SLIT2, ROBO1 significantly reduced the tumour burden induced by SW-1990 cells in subcutaneous PDAC xenograft mice (Fig. 6n, Supplementary Fig. 15e–f). In particular, IF showed that *ΔROBO1*-expressing SW-1990 cells triggered more severe apoptosis than observed in

**Fig. 4 | The SLIT2-ROBO1 axis facilitates metastasis in the liver.**
**a–c** Representative CT combined with 3D organ reconstruction bioluminescence imaging to assess liver metastasis modelled by Kpc1199$^{Robo1\text{-}FL}$ (**b**) or Panc02$^{Robo1\text{-}FL}$ (**c**) cells in *Slit2$^{fl/fl}$* or *Slit2$^{fl/fl}$/Alb*-Cre mice with or without Lenti-*loxp-Slit2* injection: Premetastatic niche (PMN) (upper) and macrometastatic niche (MMN) (lower) (*n* = 5 mice per group, mean ± SEM.; two-tailed unpaired Student's *t* test). Scale colour bars: $2.00 \times 10^5$–$2.00 \times 10^7$. **d, e** Representative CT combined with 3D organ reconstruction bioluminescence imaging displaying Kpc1199$^{Robo1\text{-}FL}$ (upper) and Panc02$^{Robo1\text{-}FL}$ (lower) cell-injected liver metastasis mouse models administered IgG or ROBO1 neutralizing antibody (*n* = 5 mice per group, mean ± SEM.; two-tailed unpaired *t* test). Scale colour bar: $2.00 \times 10^5$–$2.00 \times 10^7$. Red, reconstructed liver; orange, reconstructed spleen; yellow, signals of tumour niches. (**f–h**) The administration of IgG or ROBO1 neutralizing antibody to the *Kras$^{G12D/+}$/Trp53$^{R172H/+}$/Pdx1*-Cre (KPC) mouse model from week 12 is shown. The number of metastatic niches in the liver with diameters over or under 500 μm in each group was determined (**g**), and the percentage of ROBO1$^+$ metastatic niches in each group was assessed (**h**) (*n* = 5 mice per group, mean ± SEM.; two-tailed unpaired *t* test). (**i**) IHC-P staining of SLIT2 and ROBO1 in serial sections of livers with metastasis from KPC mice with or without ROBO1 neutralizing antibody treatment (*n* = 5 mice per group, 3 fields assessed per sample). Scale bar, 200 μm. (**j–l**) IHC-P staining of SLIT2 and ROBO1 in serial sections of livers with metastasis from KPC mice that underwent ROBO1 neutralizing antibody treatment starting from PMN (week 12-) or MMN (week 16-) occurrence. The number of metastatic niches in the liver with diameters over or under 500 μm in each group was determined (**k**), and the percentage of ROBO1$^+$ metastatic niches in each group was assessed (**l**) (*n* = 5 mice per group, mean ± SEM.; two-tailed unpaired *t* test). (**m**) IHC-P staining of SLIT2 in serial sections of KPC liver metastases that experienced ROBO1 neutralizing antibody treatment starting from PMN (week 12) or MMN (week 16) occurrence (*n* = 5 mice per group, 3 fields assessed per sample). Scale bar, 200 μm. Source data are provided in the Source Data file.

the subcutaneous tumours in the other two groups (Supplementary Fig. 15g–i).

Taken together, these data indicate that ROBO1, recognized as a DR, can lead to cell apoptosis without ligand binding while facilitating cell growth when ligands are present. This dual function of ROBO1 lays a mechanistic foundation for its selective pressure in metastatic tumour-host coadaptation.

## Activated ROBO1 facilitates cell growth by enhancing the MEK3/6-p38α MAPK interaction

Given that the mitogen-activated protein kinase (MAPK) pathway controls cell behaviours by modulating cell proliferation, migration, survival and apoptosis, we wondered whether the MAPK pathway is involved in the SLIT2-ROBO1 axis-triggered survival signal for PDAC cell metastasis to the liver[36,37]. The results revealed that the binding of rSLIT2 to ROBO1-FL significantly induced the phosphorylation of p38α MAPK but exerted little effect on other key molecules involved in this pathway. Intriguingly, the phosphorylation levels of MEK3 and MEK6, p38αMAPK-specific kinases, were not altered after rSLIT2 stimulation (Supplementary Fig. 16a). Furthermore, PH-797804 and VX-702, specific inhibitors targeting p38αMAPK, efficiently abolished the SLIT2-ROBO1-mediated cell growth advantage (Fig. 7a, Supplementary Fig. 16b). Additionally, VX-702 efficiently decreased the viability of SW-1990$^{CTRL}$ cells, and rSLIT2 stimulated SW-1990$^{ROBO1\text{-}FL}$ cells to the same level, indicating that the SLIT2-ROBO1 axis-induced growth advantage is mediated by p38αMAPK (Fig. 7b, Supplementary Fig. 16c). Since creative studies have demonstrating that p38α MAPK is vital in PMN formation and that p38 MAPKs can be rapidly activated by SLIT2-containing conditioned medium in 5 min in Xenopus retinal growth cones, we hypothesized that the p38αMAPK pathway is closely related to SLIT2-ROBO1-mediated coadaptation in the PMN and MMN[38,39]. Consistent with a previous report, our results showed that 5-10 min of rSLIT2 treatment elevated the phosphorylation level of p38αMAPK, which peaked at 30 min (Fig. 7c). This phenomenon was most obvious in PANC-1 cells with relatively high expression of ROBO1 (Fig. 7d). We further demonstrated that ROBO1 mediated rSLIT2-induced phosphorylation of p38αMAPK in SW-1990$^{ROBO1\text{-}FL}$ cells (Fig. 7e). IF staining was performed to further confirm the increase in P-p38 and nuclear translocation induced by rSLIT2 (Fig. 7f, Supplementary Fig. 16d), while the level of P-MEK3/6 was almost unchanged (Fig. 7c, d, Supplementary Fig. 16d). Furthermore, patient-derived metastatic liver specimens were used for IHC-P staining of P-p38, and the results illustrated that tumour cells with activated P-p38 were surrounded by an abundance of SLIT2 (Supplementary Fig. 16e). Then, through ex vivo testing, we proceeded to further seek direct evidence that p38αMAPK is phosphorylated in the presence of SLIT2. Mouse livers with metastatic Panc02$^{Robo1\text{-}FL}$ cells were separated and sliced into two pieces for exposure in medium with or without rSLIT2 for 1 h. The results showed more significant phosphorylation of p38αMAPK and nuclear

accumulation in the rSLIT2-treated group than in the control group (Fig. 7g, Supplementary Fig. 16f). Moreover, livers from KPC mice bearing metastases were also obtained ex vivo, and the data showed that a neutralizing antibody against ROBO1 sufficiently prevented p38αMAPK phosphorylation and nuclear translocation (Fig. 7h; Supplementary Fig. 16g).

Together, these data indicate that the phosphorylation of p38α MAPK triggered by SLIT2-ROBO1 is rapid and less sensitive to alteration of MEK3/6, which inspired us to speculate that there could be direct interactions between activated ROBO1 and p38α MAPK or MEK3/6. Co-IP revealed that ROBO1 directly bound to p38αMAPK and MEK3/6 (Fig. 7i, j). Further study confirmed that binding of SLIT2 enhanced the phosphorylation of p38αMAPK precipitated by ROBO1 (Fig. 7k). Together, these results suggest that SLIT2-bound ROBO1 can increase the opportunity for the interaction of p38αMAPK with its kinase MEK3/6 to activate the downstream pathway.

Our study unveiled a mechanism of "seed" and "soil" coadaptation, which is driven by DR-derived selection pressure through the p38αMAPK pathway.

## Discussion

Numerous DTCs derived from primary tumours are released into circulation as "seeds" for distant metastasis, but only a few of them achieve an MMN in target organs. It is also unclear why different types of tumour cells prefer specific target organs[5]. It is reasonable that DTCs and the "soil" (native microenvironment in the target organ) can reciprocally interact to achieve TMN formation. The requirements for this process are the alterability of TMN cells and the adaptability of DTCs. In our study, we revealed that hepatocytes provide SLIT2 for PMN formation only for ROBO1-positive PDAC DTCs, supporting survival at their arrival. The implantation and outgrowth of metastatic niches also stimulates more hepatocytes to provide sustained supplementation with SLIT2.

PMNs are formed by aberrantly activated native cells, such as tumour-associated macrophages (TAMs), cancer-associated fibroblasts (CAFs) and some organ-specific cells, such as hepatocytes[6,7,22,27,40]. These PMN cells exert metastasis-supportive effects by producing extracellular matrix (ECM) proteins to form niches for DTCs. Well-established definitive PMN detection strategies are lacking because PMNs are usually generated without histological alteration, and different cancer types or different distant metastases formed by the same cancer type require different PMN proteins[5]. It is also difficult to obtain clinical samples containing PMNs due to the reasons mentioned before; for example, it is often possible that PDAC patients bearing hepatic PMNs may have seemingly healthy livers diagnosed by known tests. However, intrasplenic injection mouse models can generate liver metastases at relatively consistent time points once other experimental conditions (e.g., types and amounts of injected tumour cells) are kept stable, making it possible for us to decide the approximate time point

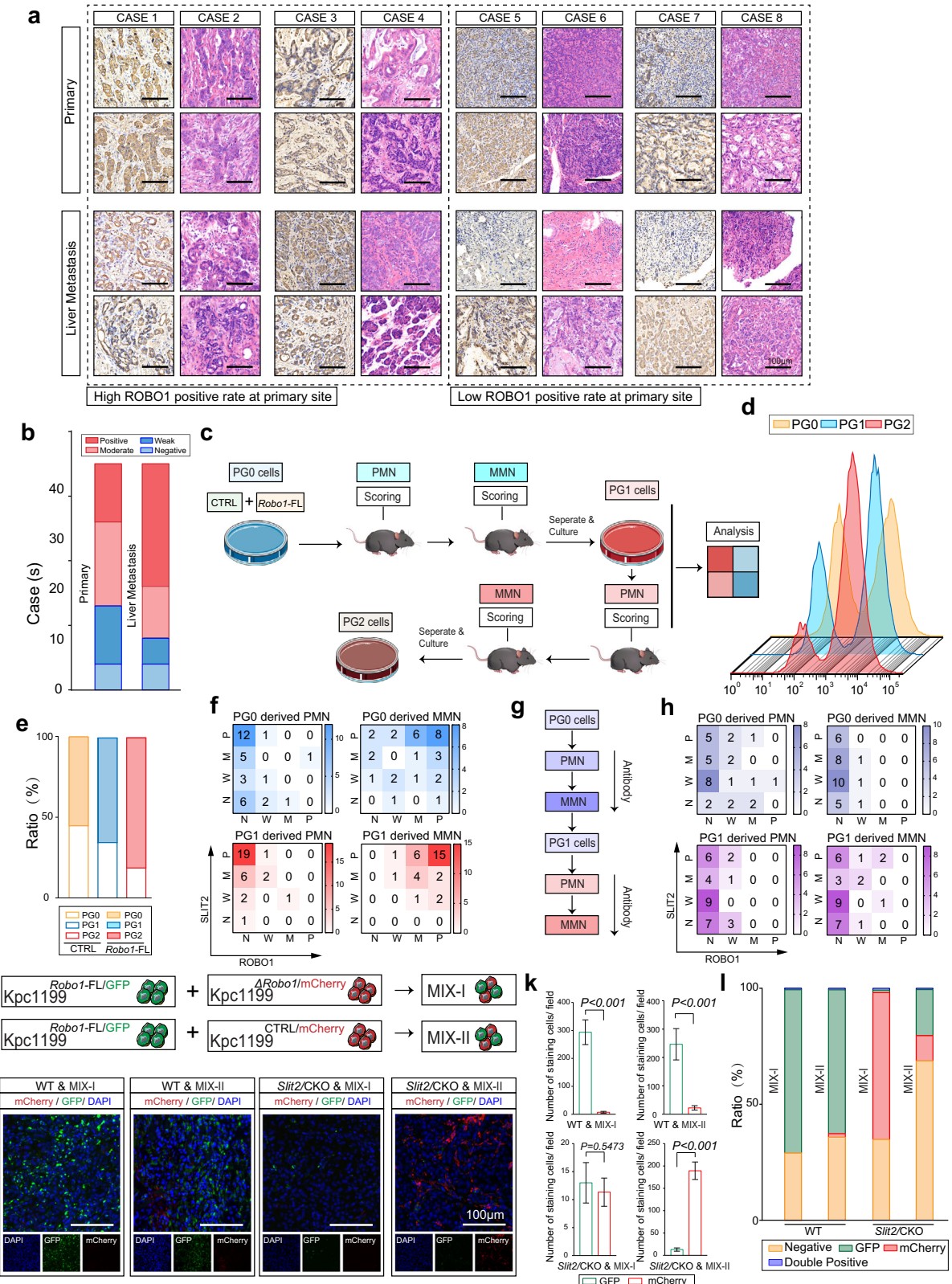

of PMN formation. Moreover, we examined tissues by utilizing reported PMN markers to ensure reliability before further explorations were continued. Finally, we determined that hepatocyte-derived *Slit2*, a member of the AG family, was enriched in the PMN and MMN, allowing tumour cells to thrive.

Receptors have long been defined as switches that can fulfil their functions only if triggered by their ligands. However, DRs can

generate and conduct contrasting signals according to the presence or absence of their ligands[15]. Furthermore, it is interesting to note that many known DRs are encoded by neuroendocrine or development-related genes and share some common characteristics; for example, their ligands are usually only abundant in specific organs[14,16,18,19]. The hypothesis postulates that DRs, which are regarded as important guardians of tissue homeostasis, perform as a

**Fig. 5 | Coadaptation is mediated by the SLIT2-ROBO1 axis. a** Representative H&E staining and IHC-P staining of ROBO1 in human PDAC primary tumours and paired liver metastatic niches displaying the importance of ROBO1 in liver metastasis ($n = 35$ cases, 3 fields assessed per sample). Scale bars, 100 μm. **b** The number of PDAC primary tumours or liver metastases expressing different ROBO1 levels according to scores on IHC-P staining is shown ($n = 35$ cases). **c** Equal amounts of Panc02$^{Ctrl}$ and Panc02$^{Robo1\text{-}FL/GFP}$ cells were mixed to prepare an intrasplenic injection model followed by metastatic niche separation and reculture. IHC-P and flow cytometry were performed to evaluate coadaptation. PG0: original cell mixture; PG1: cells derived from PG0-modelled liver metastases; PG2: cells derived from PG1-modelled liver metastases. **d–e** Flow cytometry to detect the cell composition of PG0, PG1 or PG2 ($n = 5$ technical repeats per group). **f** Heatmap showing the distribution of ROBO1 and SLIT2 expression in the livers of Panc02 cell mixture-modelled mice measured by IHC-P staining ($n = 32$ mice per group, 3 fields assessed per sample; independent experiments for each group). N: negative; W: weak; M,

moderate; P, positive. **g, h** Mice treated with ROBO1 neutralizing antibody were modelled with a Panc02 cell mixture before IHC-P staining was performed ($n = 32$ mice per group, 3 fields assessed per sample; independent experiments for each group). N: negative; W, weak; M, moderate; P, positive. **i** The strategy for exploring the coadaptation mechanism mediated by SLIT2-ROBO1 in liver metastatic niches. A mixture of equal amounts of Kpc1199$^{Robo1\text{-}FL/GFP}$ cells and Kpc1199$^{\Delta Robo1/mCherry}$ cells (Mix-I) or Kpc1199$^{Robo1\text{-}FL/GFP}$ cells and Kpc1199$^{CTRL/mCherry}$ cells (Mix-II) was intrasplenically injected into *Slit2*/CKO or CTRL mice for further examination. **j, k** Representative IF staining of a Kpc1199 cell mixture showing the formation of liver metastatic niches indicating coadaptation ($n = 5$ mice per group, 3 fields assessed per sample, mean ± SEM.; two-tailed unpaired $t$ test). GFP, green; mCherry, red; DAPI, blue. Scale bars, 100 μm. **l** Flow cytometry detection of the ratio of two types of Kpc1199 cells in separated liver metastatic niches. Source data are provided in the Source Data file.

safe lock against heterogeneous cell implantation, including cancer metastasis to other organs[15]. Under normal conditions, DRs are used to restrict cells in the given organ with abundant ligands and prevent cell outgrowth into an alien microenvironment without ligands by inducing cell apoptosis. In pathological settings, such as metastatic cancer, it is vital for DRs to exert suppressive effects on tumour cells to inhibit metastasis. Here, we revealed that PDAC tumour cells utilize this mechanism for their colonization and outgrowth in the liver by inducing hepatocytes to secrete SLIT2 from the formation of the PMN to the MMN in liver metastasis. Metastasis requires changes in both "seed" and "soil" to cater to the needs of each and eliminate unqualified cells, termed coadaptation. In this study, we tried to explain that the selective power or driving power involved in coadaptation was provided by the DR properties of ROBO1. The SLIT2-rich liver microenvironment not only supported the survival and outgrowth of disseminated ROBO1$^+$ tumour cells but also exerted selective pressure on DTCs to enrich ROBO1$^+$ cells (Fig. 8a).

Furthermore, an antibody against the SLIT2-ROBO1 axis provided a superior therapeutic effect, since blocking SLIT2 binding to ROBO1 not only weakened the proliferation advantage brought by the ROBO1-p38MAPK pathway but also triggered ROBO1-induced cell apoptosis, indicating that a therapeutic strategy involving DRs might have potential application value (Fig. 8b). Considering that surgery is not an option for PDAC patients with liver metastasis, which limits effective treatment, our study proposes a promising treatment strategy for these cases: targeting the SLIT2-ROBO1 axis. For further study, it would be interesting to explore whether this is a common mechanism in cancer metastasis.

## Methods

All the study methods in this research were approved by the Ethics Committee of Shanghai Cancer Institute, Ren Ji Hospital, School of Medicine, Shanghai Jiao Tong University.

### Constructs & reagents

Antibodies against the following proteins were purchased from Abcam: ROBO1 (C-terminal) (ab7279), SLIT2 (ab134166), ROBO2 (ab75014), ROBO3 (ab229722), TIMP1 (ab109125), LOX (ab174316), MIF (ab7027), SAA1/2 (ab199030), CD163 (ab182422), CD68 (ab955), CD11b (ab133357), Ly-6G (ab25377), keratin19 (rabbit-derived) (ab52625), keratin19 (mouse-derived), (ab7754), N-cadherin (ab76011), E-cadherin (ab1416), RAC1 + Cdc42 (phospho S71) (ab76535), P-p38 (T180 + Y182) (ab4822), P-MEK3 (S189/T193) + P-MEK6 (S207/T211) (ab4759), MEK3 + MEK6 (ab200831), active Caspase 3 (ab2302), albumin (ab207327), donkey anti-goat IgG H&L (HRP) (ab6885), Goat Anti-Mouse IgG H&L (HRP)(ab6789), Goat Anti-Rabbit IgG H&L (HRP) (ab6721) and rabbit IgG (ab172730). Antibodies against the following proteins were purchased from Cell Signaling Technology: DAPK1

(3008), active Caspase 7 (9491), active Caspase 8 (9748), JNK (9252), P-JNK (Thr183/Tyr185) (9251), ERK1/2 (4695), P-ERK1/2 (T202/Y204) (4370), c-jun (9165), P-c-jun (9261), GSK3α/β (5676), P-GSK3α/β (Y216/Y279), and mouse IgG (37988). P38α MAPK antibody (orb540329) was purchased from Biorbyt. A neutralizing antibody against the Ig1-Ig2 domain of ROBO1 (HAM1H6-1-8) and negative control IgG were purchased from HuaBio. A Duolink® proximity ligation assay (Olink Bioscience, DUO92007) kit and polybrene (H9268) were purchased from Sigma–Aldrich. The p38α MAPK-specific inhibitors VX-702 (S6005) and PH-797804 (S2726) were purchased from Selleck. Cell Counting Kit-8 (CCK-8) was purchased from Dojindo Molecular Technologies. A Caspase-3/7 Activity Kit (G7790), D-luciferin (P1043) and FuGENE transfection reagent were purchased from Promega. Puromycin (A1113802) was purchased from Gibco. Lipofectamine 2000 was purchased from Invitrogen. G-Dynabeads (10004D) were purchased from Life Technologies. Cell lysis buffer for western blotting and IP (P70100) was purchased from New Cell & Molecular Biotech. An In Situ Cell Death Fluorescein Kit TUNEL (11684795910) and SYBR Premix Ex Taq (04,913,914,001) were purchased from Roche. TRIzol reagent (9109) and a PrimeScript RT–PCR kit (RR037A) were purchased from Takara. A tumour dissociation kit (mouse) (130-096-730) was purchased from Mltienyi Biotec. RNAscope Probe SLIT2 (slit guidance ligand 2) (Cat No. 466221-C2), RNAscope Probe KRT19 (keratin 19) (Cat No. 310221-C1), RNAscope Probe ALB (albumin) (Cat No.600941-C1), RNAscope Probe ACTA1 (actin alpha 1, skeletal muscle) (Cat No.454319-C1) and RNAscope® Multiplex Fluorescent Reagent Kit (Cat No. 323100) were purchased from ACDbio.

### Clinical samples

The specimens examined in this study mainly included 2 cohorts for respective experimental designs: Cohort I, containing 35 cases of liver metastasis tissues from PDAC patients together with paired primary tumour tissues, was used for IHC-P or IF analysis; cohort II, containing 266 cases of PDAC primary tumour tissue from patients with their respective prognosis, was used for IHC-P and survival analysis (Supplementary Table 2).

All cases of PDAC or PDAC liver metastases were obtained from Ren Ji hospital from February 2004 to September 2014. All patients had not received radiotherapy, chemotherapy, hormone therapy or other related anti-tumor therapies before surgery.

The study was approved by the Research Ethics Committee of Ren Ji Hospital, School of Medicine, Shanghai Jiao Tong University. Written informed consent was provided before enrollment. Approval letter of Shanghai Jiaotong University School of Medicine, Renji Hospital Ethics Committee is RA-2019-116.1.

### Animal experiments

lox-stop-lox-*Kras*$^{G12D/+}$; lox-stop-lox-*Trp53*$^{R172H/+}$; *Pdx1*-Cre mice were purchased from The Jackson Laboratories (Bar Harbor, ME) to generate

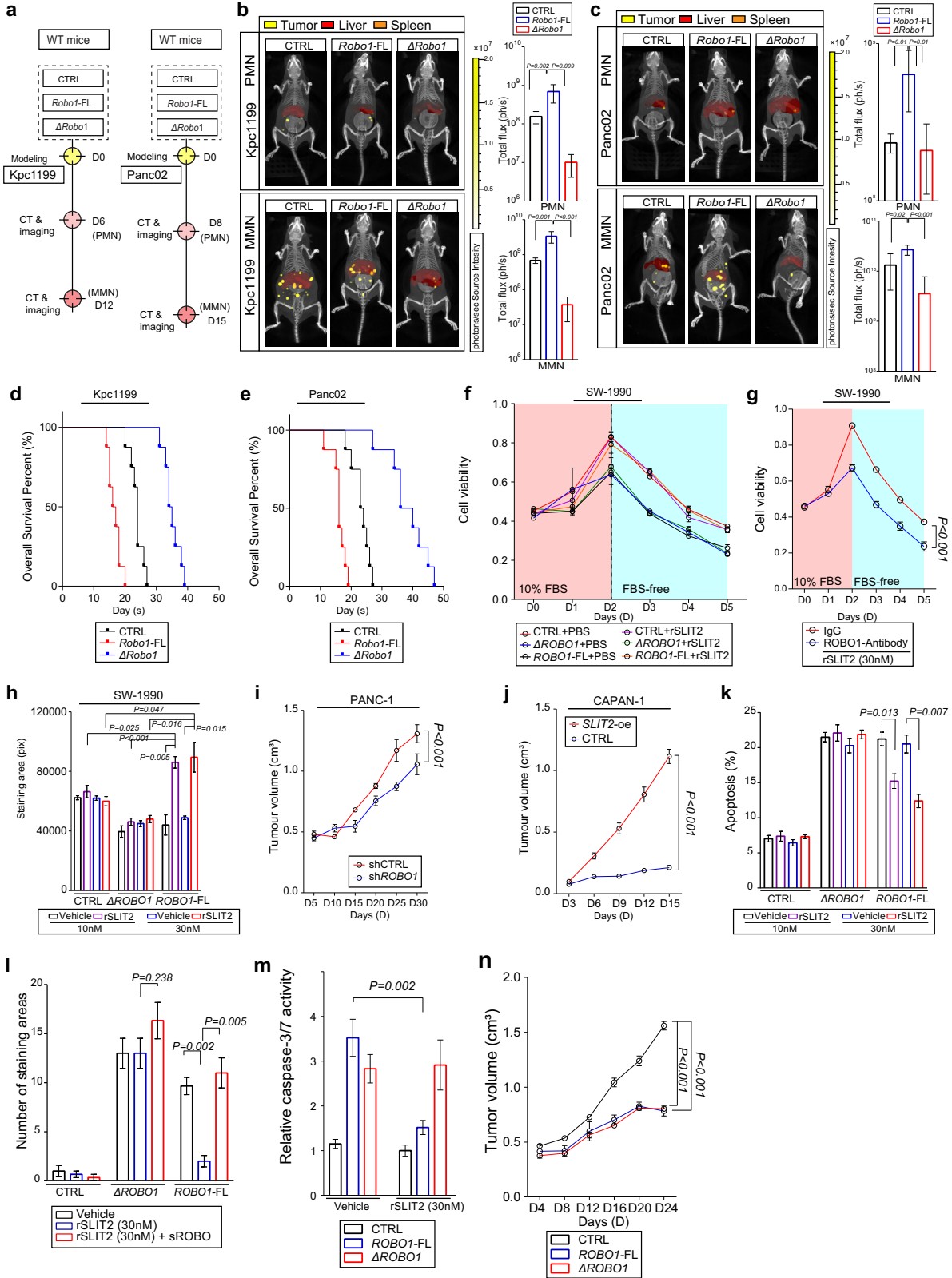

a transgenic PDAC mouse model. *Slit2*[fl/fl] mice and *Alb1*-Cre mice were purchased from Cyagen. All the above mice were on the C57BL/6 genetic background. All C57BL/6 wild-type mice and nu/nu mice were purchased from East China Normal University.

16-24 weeks old KPC mice and 8 weeks old WT or Slit2 KO mice or nude mice were performed in this study. All mice above were on the C57BL/6 genetic background.

Animals were housed in East China Normal University SPF animal facility, in temperatures 20–22 °C, humidity 30–70% and a 12-h light/12-h dark cycle. All animals received humane care according to the criteria outlined in the "Guide for the Care and Use of Laboratory Animals" prepared by the National Academy of Sciences and published by the National Institutes of Health. All manipulations were performed under approved protocol number

**Fig. 6 | ROBO1 exhibits the characteristics of a dependence receptor.**
**a**–**c** Representative CT combined with 3D organ reconstruction bioluminescence imaging displaying the outgrowth ability of Kpc1199 cells (**b**) or Panc02 cells (**c**) in mouse models: Premetastatic niche (PMN) (upper) and macrometastatic niche (MMN) (lower) ($n = 5$ mice per group, mean ± SEM.; two-tailed unpaired $t$ test). Scale colour bars: $2.00 \times 10^5$–$2.00 \times 10^7$. **d, e** Survival analysis of intrasplenic mouse models bearing Kpc1199 or Panc02 injection (**e**) ($n = 8$ mice per group). **f** Viability of SW-1990 cells with (+rSLIT2) or without (+PBS) 30 nM rSLIT2 exposure ($n = 2$ biological replicates, mean ± SEM., one-way repeated-measures ANOVA). The Y-axis represents OD values at 450 nm. (**g**) Viability of SW-1990$^{ROBO1\text{-FL}}$ cells exposed to 30 nM rSLIT2 with or without ROBO1 neutralizing antibody ($n = 2$ biological replicates, mean ± SEM., one-way repeated-measures ANOVA). The Y-axis represents OD values at 450 nm. (**h**) Colony formation assays evaluating the outgrowth ability of SW-1990 cells expressing ROBO1-FL or ΔROBO1 exposed to 10 nM or 30 nM rSLIT2 ($n = 2$ biological replicates, mean ± SEM.; two-tailed unpaired $t$ test); (**i**) Tumour sizes in subcutaneous xenograft models utilizing PANC-1$^{shCTRL}$ or PANC-1$^{shROBO1}$ cells ($n = 2$ biological replicates, $n = 7$ mice per group, tumour volume was calculated as volume = $0.5 \times$ length $\times$ width2, mean ± SEM., one-way Repeat-Measure ANOVA). **j** Tumour growth in subcutaneous xenograft models utilizing CAPAN-1$^{CTRL}$ or CAPAN-1$^{SLIT2\text{-oe}}$ cells ($n = 2$ biological replicates, $n = 6$ mice per group, tumour volume was calculated as volume = $0.5 \times$ length $\times$ width2, mean ± SEM., one-way repeated-measures ANOVA). **k** Apoptosis of SW-1990 measured by flow cytometry with dual PI and Annexin V staining ($n = 2$ biological replicates, $n = 3$ tests per group, mean ± SEM.; two-tailed unpaired $t$ test). (**l**) TUNEL assays performed on SW-1990$^{CTRL}$, SW-1990$^{\Delta ROBO1}$ and SW-1990$^{ROBO1\text{-FL}}$ cells with or without 30 nM rSLIT2 exposure ($n = 3$ biological replicates, 3 fields assessed per sample, mean ± SEM., two-tailed unpaired $t$ test). TUNEL staining, green; DAPI, blue. Scale bars: 50 μm. (**m**) Caspase-3/7 activity in SW-1990 cells measured 48 h after serum starvation ($n = 3$ biological replicates, $n = 5$ tests per group, mean ± SEM., two-tailed unpaired $t$ test). **n** Tumour growth in subcutaneous xenograft models utilizing SW-1990$^{CTRL}$, SW-1990$^{\Delta ROBO1}$ and SW-1990$^{ROBO1\text{-FL}}$ cells measured every 4 days ($n = 6$ mice per group, tumour volume was calculated as volume = $0.5 \times$ length $\times$ width2, mean ± SEM., one-way Repeat-Measure ANOVA). Source data are provided in the Source Data file.

---

20141204 assigned by the Research Ethics Committee of East China Normal University.

We confirmed that the maximal tumour size/burden was not exceeded 200 mm³ during our study according to institutional review board. Investigators conducting these animal experiments were blinded to allocation during experiments and outcome assessments.

## Liver metastasis model and definition of PMN and MMN
The intrasplenic injection model provides repeatable and controllable liver metastasis at a relatively fixed time in mice, which makes research on the PMN and MMN possible[7,25]. The disadvantages of this model are that the progression of liver metastasis is more rapid than that in spontaneous models, such as the KPC model, while liver metastasis is generated at a stable location and time point when the same experimental conditions are applied[21].

First, we explored the time points at which MMNs formed in most modelled mice using Kpc1199 cells or Panc02 cells at different concentrations through IHC-P in liver left lobule sections every 3 days. We then chose Kpc1199 cells at $4 \times 10^5$/mouse or Panc02 cells at $1 \times 10^6$/mouse to ensure that the formation of MMNs would not be too early or late, which would make it difficult to induce PMN formation at consistent time points. The results of IHC-P and CT combined with 3D organ reconstruction bioluminescence imaging finally defined the approximate time point of MMN formation (Day 12 for Kpc1199 cells and Day 9 for Panc02 cells). In preliminary data involving 50 mice for each cell line, more than 90% of mice generated MMNs at the right time (98% for Kpc1199 cells on Day 12 and 92% for Panc02 cells on Day 9). Then, we examined liver sections before MMN formation by visualizing GFP and staining for the reported PMN markers LOX, TIMP-1, MIF, and SAA1/2 and markers of M2-type macrophages. PMNs recognized as GFP⁻/markers⁺ liver sections were then determined (Days 7-11 for Kpc1199 cells and Days 5-8 for Panc02 cells). In preliminary data obtained from 50 mice for each cell line, more than 85% of the mice generated PMNs at the right time according to the previously mentioned markers (90% for Kpc1199 cells on Day 8 and 86% for Panc02 cells on Day 6). We thus identified the time point of both of these events (Fig. 1b; Supplementary Fig. 1a).

In brief, $4 \times 10^5$ Kpc1199 cells, $1 \times 10^6$ Panc02 cells or $1 \times 10^6$ LTPA cells suspended in 20 μl DMEM without FBS were used. For PANC1 cells modelled mouse, $1 \times 10^6$ PANC1 suspended in 20 μl DMEM without FBS were used to perform nude mice injection. In immunocompetent isogenic C57BL/6 or nude mice under 2.5% isoflurane inhalation anaesthesia, a 10-15 mm subcostal incision was made through the abdominal skin and peritoneum for surgical exposure of the spleen after sterilization. The speed of injection was slower than 5 μl/s, followed by 2 s of needle retention to prevent leakage. Wound closure was then rapidly performed. For Sham group, only PBS was injected.

For retrieval of expression of SLIT2 in SLIT2 conditional knockout mice, we used the vector GV348 (Ubi-MCS-SV40-puromycin) containing loxp-NM_004787-loxp as the lentivirus package. For efficiency examination, NM_004787-GFP fusion was carried.

Then, 200 μl saline with or without a dose of $3 \times 10^7$ HIV was delivered into the tail vein of the mice. Mice were anaesthetized with 2.5% vapourized inhaled isoflurane and placed in a restraint that positioned the mouse tail in a lighted, heated groove. The speed of injection was slower than 50 μl/s. The efficiency of restoration and performance were evaluated 10 days later. Treatment was performed 2 weeks before the intrasplenic injection of tumour cells.

For neutralizing antibody treatment, an antibody against the Ig1-Ig2 domain of ROBO1 was intraperitoneally injected 1 week before hepatic metastasis modelling, and the antibody (500 μg/mouse) was administered every 3 days until the mice were sacrificed. IgG was performed as negative control. For administration of neutralizing antibody to KPC mice, intraperitoneal injection was performed every 3 days from week 12 or week 16 until week 18 before tissue harvesting.

## Subcutaneous Xenograft Model
Athymic male nu/nu mice aged 6 to 8 weeks were used for the subcutaneous xenograft model. For SW-1990 cells, SW-1990$^{CTRL}$, SW-1990$^{\Delta ROBO1}$ and SW-1990$^{ROBO1\text{-FL}}$ cells suspended in DMEM at a concentration of $1 \times 10^7$ cells/ml were injected, the injection volume was 200 μl, and tumour diameters were monitored with callipers every 4 days until the sacrifice of mice on Day 24; for CAPAN-1 cells, CAPAN-1$^{CTRL}$ and CAPAN-1$^{SLIT2\text{-oe}}$ cells suspended in DMEM at a concentration of $1 \times 10^7$ cells/ml were injected, the injection volume was 150 μl, and tumour diameters were monitored with callipers every 3 days until the sacrifice of mice on Day 15; for PANC-1 cells, PANC-1$^{shCTRL}$ and PANC-1$^{shROBO1}$ cells suspended in DMEM at a concentration of $2 \times 10^7$ cells/ml were injected, the injection volume was 200 μl, and tumour diameters were monitored with callipers every 5 days until the sacrifice of mice on Day 40. Inoculation was performed under the right inguinal skin of mice. Tumour volumes were calculated as volume = $0.5 \times$ length $\times$ width$^2$. After the mice were sacrificed, the tumours were separated from the body, and their weights were measured.

## CT combined with 3D organ reconstruction bioluminescence imaging
Mice bearing hepatic metastases composed of luciferase-expressing tumour cells were intraperitoneally injected with 150 mg D-luciferin at a volume of 200 μl. Mice were then anaesthetized with 2.5% vapourized inhaled isoflurane 2 min after injection before being placed into an IVIS Spectrum imaging system (Caliper Life Sciences, Waltham, MA). CT was then performed to merge with firefly bioluminescence signals. For organ reconstruction, CT images were placed combined

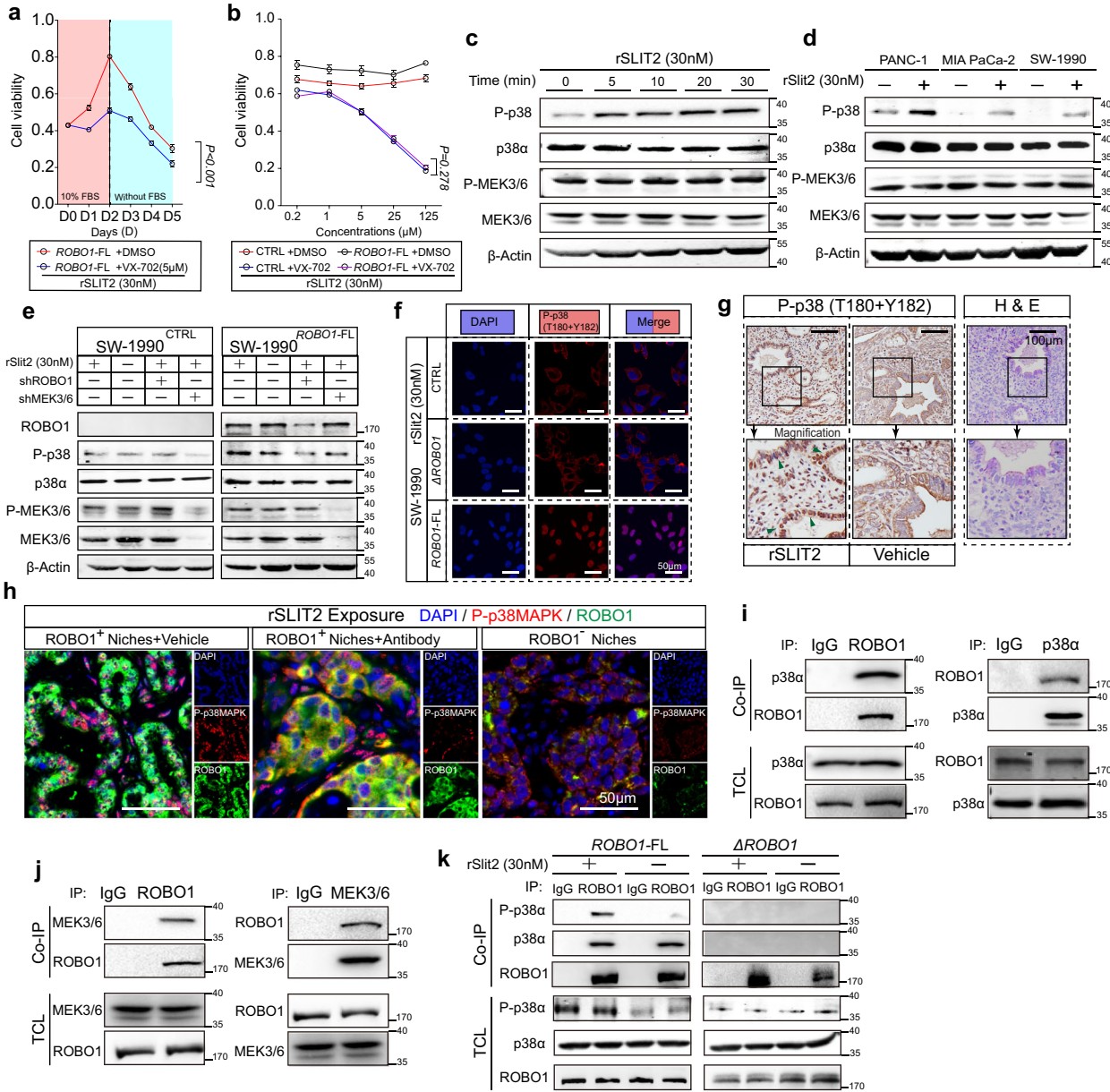

**Fig. 7 | The SLIT2-ROBO1 axis triggers p38αMAPK. a** Viability of SW-1990^ROBO1-FL cells exposed to 30 nM rSLIT2 with or without the p38αMAPK-specific inhibitor VX-702 (5 μM) (n = 2 biological replicates, mean ± SEM., one-way repeated-measures ANOVA). The Y-axis represents OD values at 450 nm. **b** Viability of SW-1990^CTRL and SW-1990^ROBO1-FL cells treated with 30 nM rSLIT2 with or without the p38αMAPK-specific inhibitor VX-702 at various concentrations (0.2 μM, 1 μM, 5 μM, 25 μM, 125 μM; time point = D3; n = 2 biological replicates; mean ± SEM.; one-way Repeat-Measure ANOVA). *ns. no significant difference, P > 0.05*. The Y-axis represents OD values at 450 nm. **c** WB displaying the time-dependent rSLIT2-induced p38αMAPK phosphorylation in SW-1990^Robo1-FL cells (n = 2 biological replicates). **d** WB showing the phosphorylation of p38αMAPK in PANC-1, MIA PaCa-2 or SW-1990 cell lines treated with 30 nM rSLIT2 (n = 2 biological replicates). **e** WB displaying the relationship of SLIT2 (30 nM), ROBO1 and MEK3/6 expression with phosphorylation of MEK3/6 or p38αMAPK (n = 2 biological replicates). **f** Representative ICC staining in 30 nM rSLIT2-treated SW-1990^CTRL, SW-1990^ΔROBO1 and SW-1990^ROBO1-FL cells

displaying the nuclear translocation of p38αMAPK (3 fields assessed per sample). P-p38MAPK, red; DAPI, blue. Scale bars: 50 μm (n = 2 biological replicates). **g** Representative ex vivo IHC-P staining of P-p38 showing rSLIT2-induced phosphorylation of p38αMAPK in cultured Panc02^Robo1-FL cells from metastases in mouse livers (n = 6 samples per group, 3 fields assessed per sample). Scale bars, 100 μm. (**h**) Representative ex vivo IF staining of P-p38 and ROBO1 in Kras^G12D/+/Trp53^R172H/+/Pdx1-Cre (KPC) mouse model metastatic niches formed in livers with or without ROBO1 neutralizing antibody treatment. ROBO1, green; P-p38MAPK, red; DAPI, blue (n = 3 mice per group, 3 fields assessed per sample). Scale bar, 50 μm. **I, j** Lysate of the PANC-1 cell line endogenously immunoprecipitated with anti-ROBO1 and anti-p38α MAPK (**i**) or anti-ROBO1 and anti-MEK3/6 (**j**) antibodies and immunoblotted with the indicated antibodies (n = 2 biological replicates). **k** Lysate of rSLIT2 (30 nM)-induced SW-1990 cells immunoprecipitated with anti-ROBO1 antibody and immunoblotted with the indicated antibodies (n = 2 biological replicates). Source data are provided in the Source Data file.

with the autoreconstructed liver and spleen images at an opacity of 20%. The normalization and quantification of firefly bioluminescence signals were performed according to the surrounding area of the red cube. Mice were sacrificed at the end of the test, and the tumour-bearing livers were immediately isolated and then washed in ice-cold PBS. A photo of luciferin emission imaging was then taken of each liver. All imaging and calculations were performed with Living Image software, version 4.5.3.

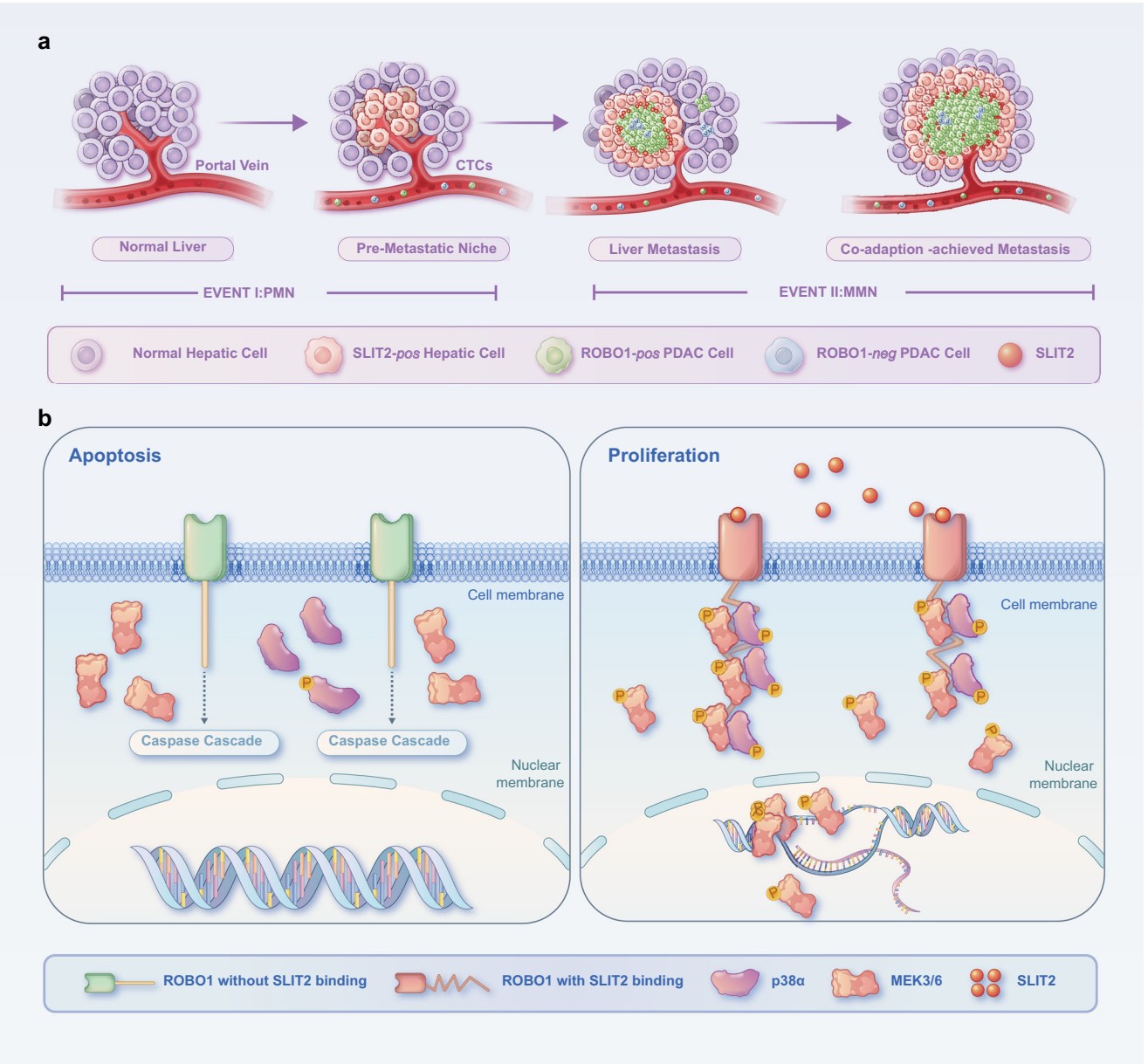

**Fig. 8 | Summary model displaying the mechanism of the SLIT2-ROBO1 axis in PDAC liver metastasis and coadaptation.** (a) Summary model displaying the process of SLIT2-ROBO1-mediated coadaptation of hepatic cells and tumour cells, which promotes metastatic niche outgrowth in PDAC. **b** Summary model showing the dual roles of ROBO1 in coadaptation with or without SLIT2. ROBO1 induces apoptosis through the caspase cascade in the absence of SLIT2 and triggers proliferation and survival through the p38MAPK pathway by interacting with SLIT2.

## Transcriptional analysis

Mice were intrasplenically injected with Panc02 or Kpc1199 cells. Then, the mice were sacrificed on specific days, and the livers were harvested. For Panc02 model mice, the time points were Day 3, Day 6 and Day 12; for Kpc1199 model mice, the time points were Day 3, Day 8 and Day 15. The fresh livers obtained were then washed with ice-cold saline, and the left hepatic lobules were sliced into $2 \times 2 \times 2$ mm³ cubes. In particular, for tissues from Panc02 model mice obtained on Day 12 and Kpc1199 model mice obtained on Day 15, obvious metastatic niches could be observed, while only adjacent livers without metastasis were selected for further study. In the follow-up study, PMN and MMN markers were measured before transcriptional analysis. The analysis data is available in the Sequence Read Archive (SRA), the number of which is PRJNA590588. This database contains 2 parts of analysis data: Panc02 cell line modelled liver metastasis part (Panc02_D3_(1-3),Panc02_D6_(1-3) and Panc02_D12_(1-3)) and

KPC1199 cell line modelled liver metastasis part (D3_(1-3),D8_(1-3) and D15_(1-3)).

For GSEA, 4GB (64 bit) GSEA v4.01 Java Web Start (all platforms) was used. Analyses were performed as D3 vs. D8 and D3 vs. D15 for Kpc1199 cells and D3 vs. D6 and D3 vs. D12 for Panc02 cells. The gene sets involved in these analyses were "c2.cp.kegg.v7.0.symbols.gmt".

## Metastatic tumour dissociation, culture and examination

Intrasplenic injection model mice bearing liver metastases were sacrificed after the operation. The livers with metastases were then carefully obtained and immediately washed in ice-cold saline. Metastatic niches were carefully recognized and separated from the liver. Fine-pointed forceps were used for detachment of the remaining liver parenchymal tissue and tumour-associated fibrous or necrotic areas. Then, "clean" metastatic liver tumours were washed with ice-cold DMEM 3 times and cut into $1 \times 1 \times 1$ mm cubes before being transferred

into 2.5 mL DMEM containing 100 μL of Enzyme D, 50 μL of Enzyme R, and 12.5 μL of Enzyme A provided in the kit. After suspension of the dissociated tumour tissues in the previously mentioned solution, the tubes were tightly closed and placed on a constant temperature oscillator for 1 h at 37 °C and 120 rpm. The mixtures were then centrifugation at 450 × g for 5 min, and the samples were resuspended in DMEM 3 times. The suspended tumour cells in the DMEM were then counted before being seeded in dishes. Briefly, approximately 5 × 10⁴ cells in complete DMEM were added to 6-well plates for further culture. Examination of GFP or mCherry expression and luciferase activity was taken into consideration for tumour cell verification.

### Histology, immunohistochemistry and in situ hybridization, RNA-ISH (RNAscope)

Tissues were fixed in 4% paraformaldehyde and embedded in paraffinized blocks. The blocks were cut into sections at a thickness of 5 μm and then deparaffinized and rehydrated for histopathological evaluation. For haematoxylin-eosin (HE) staining, the sections were dyed in haematoxylin for 5 mins and in eosin for 5–10 s. For immunohistochemical staining, the sections were placed in sodium citrate buffer solution (pH 6.0) for 20 min for the repair of tissue antigens. Then, the sections were incubated with 0.3% hydrogen peroxide/phosphate-buffered saline for 30 mins and blocked with 10% (w/v) BSA (Sangon)/ phosphate-buffered saline (PBS). Slides were first incubated with primary antibodies at 4 °C overnight at an optimal dilution. After three washes with 1 × PBS, the slides were labelled with HRP-conjugated secondary antibody at room temperature for 1 h. Then, the sections were washed three times in 1 × PBS, treated with DAB substrate liquid (Thermo) and counterstained with haematoxylin. All sections were observed and photographed with a microscope (Carl Zeiss). Primary antibodies targeting the following proteins were used: ROBO1 (N-terminal) (1:300), ROBO2 (1:300), ROBO3 (1:500), SLIT2 (1:500), CK19 (1:500), p38α (1:700), P-p38 (T180 + Y182) (1:500), CD163(1:300);CD68 (1:100), CD11b (1:200), Ly-6G (1:300), LOX (1:500), MIF (1:300), TIMP-1 (1:500), SAA1/2 (1:700),Albumin (1:500), N-cadherin (1:500), E-cadherin (1:500) and RAC1 + Cdc42 (phospho S71) (1:500). The following secondary antibodies were utilized: HRP-conjugated anti-goat (1:1000), HRP-conjugated anti-rabbit (1:500), and HRP-conjugated anti-mouse (1:500).

For RNAscope, RNAscope® Multiplex Fluorescent Reagent Kit was used to treat paraffin-embedded specimens. All tests were performed following the manufacturer's instructions. RNAscope Probe SLIT2, RNAscope Probe KRT19, RNAscope Probe ALB (albumin), RNAscope Probe ACTA1 were used.

### Immunofluorescence staining

For tissue staining, the tissue sections were placed in pH 6.0 sodium citrate buffer solution for 20 min for heat-mediated antigen retrieval. Then, the sections were blocked with 10% (w/v) BSA/PBS and coincubated with primary antibodies at 4 °C overnight at optimal dilutions. For cell staining, SW1990 cells at 2 × 10⁴ cells per well, PANC1 cells at 2 × 10⁴ per well or MIA PaCa-2 cells at 3 × 10⁴ cells per well were seeded on slides in 12-well plates (ibidi) and cultured at 37 °C. After three washes with 1 × PBS, the slides were labelled with Alexa Fluor 594-conjugated anti-rabbit antibody and Fluor 488-conjugated anti-mouse antibody at room temperature for 1 h. Cell nuclei were stained with DAPI (Sigma) after the redundant secondary antibody was removed. After three washes with 1 × PBS, the immunofluorescence signals were captured using confocal microscopy (Carl Zeiss).

### Ex vivo living tumour tissue culture

Liver tissues containing metastatic niches formed by Panc02$^{Robo1\text{-}FL}$ cells obtained from the intrasplenic injection mouse model or liver tissue with metastases from the KPC mouse model were separated on ice to maintain their structural integrity and activity. Then, the tissues were

washed in ice-cold saline within 15 min after resection and sliced into small cubes with an approximate volume of 5 × 5 × 5 mm³. Each cube contained liver metastatic niches and adjacent liver tissue. Then, the cubes were placed into complete medium with or without 30 nM rSLIT2 administration/ROBO1 neutralizing antibody.

For intrasplenic model-derived livers, the tissues were stimulated for 1 h to measure the rapid phosphorylation of p38αMAPK. Importantly, a cube was cut into two pieces in the middle and placed into the medium with or without rSLIT2 to ensure that the same tumour microenvironment (TME) of metastatic niches was undergoing treatment. After rSLIT2 administration, all tissues were immediately fixed with 4% paraformaldehyde for further IHC-P testing.

KPC mouse-derived liver tissues were treated with antibodies for no more than 8 h to maintain the tissue constructures. All tissues were then immediately fixed with 4% paraformaldehyde followed by IF staining.

### Recombinant SLIT2 and sROBO protein expression and purification

Episomal expression vectors with pCEP-Pu-Strep II-tags were cloned into the SLIT2 or sROBO1 ORF. The 293 T cell line was transfected with the reconstructed plasmids for recombinant protein expression. Puromycin for screening was administered in complete DMEM at a concentration of 5-10 μg/ml 48 h after transfection for 10 days, and 2 μg/ml puromycin was administered to maintain the expression-positive cells. Then, the culture medium of transfected 293 T cells was collected and applied to a Strep Tactin Sepharose column (IBA, 2-1202-101), followed by washing with binding buffer and elution buffer containing 2.5 mM desthiobiotin on ice. The harvested fractions were further quantified using a Nanodrop 2000 spectrophotometer (Thermo), and western blotting was performed for identification. See also our previous study[41].

### Cell culture

Human PDAC cell lines PANC-1, BxPC-3, CFPAC-1, HPAC, CAPAN-1, CAPAN-2, Patu 8988, MIA PaCa-2, SW-1990 and AsPC-1 were purchased from ATCC; Murine cell lines Panc02, Kpc1199 and LTPA were gifting from Professor Jing Xue (State Key Laboratory of Oncogenes andRelated Genes, Shanghai Cancer Institute, Ren Ji Hospital, School of Medicine, Shanghai Jiao Tong University). All cell lines were validated using short tandem repeat (STR) profiling. All cells were cultured under the recommended conditions following granted protocols, including the respective medium supplemented with 10% (v/v) FBS and 1% antibiotics, and kept in 37 °C humidified incubators with 5% CO₂. The medium contained 4.5 g/L glucose and 2 mM L-glutamine when cellular function assays were performed.

### Cell transfection

The lentiviruses carrying shRNA sequences used for transfection with shRNA are shown in Table 1

For the negative control, scramble shRNA targeting no known genes was designed and used.

For overexpression, the pcDNA3.1-*ROBO1*-FL vector, pcDNA3.1-*ΔROBO1* vector or pcDNA3.1-*SLIT2* vector was utilized for lentivirus packaging and cell transfection. The empty vector was used as the negative control.

All transfected cells were screened with puromycin at a concentration gradient of 1 μg/ml to 10 μg/ml for at least 10 days or more before the transfection efficiencies were evaluated.

**Table 1 | Shown are shRNA sequences**

| Gene | Sequence (5' to 3') |
| --- | --- |
| sh*ROBO1* (NM_002941.4) | GGAGAGAAGGGAGTCAGAATCTACT |
| sh*MEK3* (NM_002756.4) | GCTGATGACTTGGTGACCATC |
| sh*MEK6* (NM_002758.4) | GATTTAGACTCCAAGGCTTGC |

## Cell viability assay

Cells were seeded into 96-well plates. The numbers of plated cells were as follows: for the human cell lines SW-1990, 3000/well; BxpC-3, 3000/well; PANC-1, 5000/well; CFPAC-1, 6000/well; CAPAN-1, 3000/cell; HPAC, 5000/well; and for the murine cell line Panc02, 3000/well. The measurement or treatment of these cells started after overnight incubation and was considered Day 0. The consistent measurements lasted from Day 0 to Day 5; for cell stress exertion, culture medium without FBS was added after Day 2. For rSLIT2 treatment, the concentration was 30 nM in culture medium. VX-702 treatment was performed at a concentration of 5 µM for normal cell viability assays and at a concentration gradient of 200 nM, 1 µM, 5 µM, 25 µM, and 125 µM for concentration dependence tests. PH-797804 treatment was performed at a concentration of 50 µM for the normal cell viability assay and at a concentration gradient of 80 nM, 400 nM, 2 µM, 10 µM, and 50 µM for the concentration dependence test. sROBO was added at a concentration of 5 µM in culture medium. At the indicated time points, 100 µl of diluted CCK-8 reagent in culture medium at a concentration of 10% (v/v) was added to every well and incubated with cells at 37 °C for 1 h. Cell viability was monitored using a Power Wave XS microplate reader (BIO-TEK) by measuring the absorbance at 450 nm. For each group, 5 replicate wells were prepared, and each experiment was performed three times.

## Colony formation assay

The Panc02, Kpc1199 and SW-1990 cell lines cultured at 70% confluence were utilized for colony formation assays to ensure that they were in the logarithmic growth phase. Cells were detached with 0.25% trypsin/0.01% EDTA and seeded in 3 mm dishes in complete medium with or without rSLIT2 treatment (10 nM or 30 nM) for growth. The numbers of seeded cells were as follows: SW-1990, 4000/dish; Panc02, 3000/dish; Kpc1199, 3000/dish. The medium was changed every 3 days. All cells were then fixed in 4% paraformaldehyde for 30 min and stained with crystal violet. The results were assessed using ImageJ by calculating the pixels of the staining area.

## Cell apoptosis assay

Cell apoptosis was measured using a Caspase-3/7 Activity Kit. SW-1990$^{CTRL}$, SW-1990$^{\Delta ROBO1}$ and SW-1990$^{ROBO1-FL}$ cells were plated at a concentration of 4000 cells/well into 96-well plates. Serum starvation was performed for 48 h after 3 days of growth with or without 30 nM rSLIT2 administration. Then, the activity of Caspase-3/7 was measured strictly according to the manufacturer's guidance.

Cell apoptosis was measured via Annexin V and propidium iodide (PI) staining: SW-1990$^{CTRL}$, SW-1990$^{\Delta ROBO1}$ and SW-1990$^{ROBO1-FL}$ cells were detached with 0.25% trypsin/0.01% EDTA in 1 × PBS after treatment with or without rSLIT2 administration (10 nM or 30 nM). Then, the suspended cells were harvested in DMEM and centrifuged at 400 × g for 3 min. After washing with 1 × PBS, the cells were stained with 3.5 µl Annexin V and 3.5 µl PI diluted in 100 µl binding buffer. Flow cytometry (BD) was performed for analysis after incubation of cells for 20 min at 25 °C.

Cell apoptosis was measured via a terminal deoxynucleotidyl transferase (TdT) dUTP nick-end labelling (TUNEL) assay. All steps were strictly conducted in accordance with the kit instructions.

**Quantitative real-time PCR.** Total RNA extraction was performed using TRIzol reagent followed by reverse transcription to harvest cDNA. A 7500 Real-time PCR system (Applied Biosystems, USA) was used to perform qPCR at recommended cycling settings. The conditions were 1 initial cycle at 95 °C for 2 min followed by 35 cycles of 5 sec at 95 °C and 31 sec at 60 °C. The 2-ΔCT method was utilized for evaluation and normalization to 18 S mRNA levels when relative mRNA expression was calculated.

The primers mentioned are listed in Table. 2.

**Table 2 | Shown are primer sequences used in this article**

| Gene | Forward primer (5' to 3') | Reverse primer (5' to 3') |
|---|---|---|
| ROBO1 | GACAAAACCCTTCGGATGTCA | CCAGTGGAGAGCCATCTTTCT |
| SLIT2 | GCGAAGCTATACAGGCTTGAT | TGCAGTCGAAAAGTCCTAAGTTT |
| Robo1 | GACAAAACCCTTCGGATGTCA | CCAGTGGAGAGCCATCTTTCT |
| Slit2 | CACGTGCGTCGTCCTGAAGGCT | TCTCCCAGGTAGCCAGGCAAACAC |
| Cd133 | CCTTGTGGTTCTTACGTTTGTTG | CGTTGACGACATTCTCAAGCTG |
| Cd44 | TCGATTTGAATGTAACCTGCCG | CAGTCCGGGAGATACTGTAGC |
| Krt19 | GGGGGTTCAGTACGCATTGG | GAGGACGAGGTCACGAAGC |
| 18S | TGCGAGTACTCAACACCAACA | GCATATCTTCGGCCCACA |

**Western blotting.** For cell protein extraction, cells were lysed in Cell Lysis Buffer for Western and IP (New Cell & Molecular Biotech; P70100) and Protease and Phosphatase Inhibitor Cocktail (New Cell & Molecular Biotech; P002) on ice for 10 min before centrifugation at 12,000 × g for 15 min at 4 °C. For tissue protein extraction, tissues were lysed in lysis buffer at a volume 10× the weight of tissue (g) ml on ice for 15 min before centrifugation at 13,500× g for 20 min at 4 °C; the lysis buffer contained NCM RIPA Buffer (New Cell & Molecular Biotech; WB3100) and Protease and Phosphatase Inhibitor Cocktail. The supernatants were collected, and the protein concentration was measured with a BCA Protein Assay Kit (Pierce Biotechnology) followed by standardization. Then, 5 × SDS–PAGE Sample Loading Buffer (Beyotime; P0015) was added before boiling in a water bath for 10 min. Cytoplasmic protein extraction and nuclear protein extraction were performed using NE-PER™ Nuclear and Cytoplasmic Extraction Reagents (Thermo Fisher Scientific) strictly following the guidance of the manufacturer. Protein lysates were separated via 8-12% SDS–PAGE followed by transfer onto NC membranes. For normal proteins, skimmed milk powder (Invitrogen) diluted in TBST (containing 1‰ Tween 20) at a concentration of 5% and for phosphorylated proteins, BSA (Sangon) powder diluted in TBST at a concentration of 5% were utilized to block non-specific binding for 1-2 h at room temperature. Primary antibodies targeting the following proteins were used: ROBO1 (N-Terminal) (1:500), ROBO1 (C-Terminal) (1:1000), SLIT2 (1:1000), ROBO2 (1:1000), ROBO3 (1:500), P-MEK3 (S189/T193) + P-MEK6 (S207/T211) (1:500), MEK3 + MEK6 (1:1000), p38α (1:1000), P-p38 (T180 + Y182) (1:1000), β-Actin (1:10000), GSK3α/β (1:1000), P-GSK3α/β (Y216/Y279) (1:1000), c-jun (1:1000), P-c-jun (S63) (1:1000), JNK (1:1000), P-JNK (T183/Y185), ERK1/2 (1:1000), P-ERK1/2 (T202/Y204), DAPK1(1:1000), active Caspase 7(1:1000) and active Caspase 8 (1:1000). After overnight incubation with primary antibodies diluted in Universal Antibody Diluent (New Cell & Molecular Biotech; WB500D), the membranes were probed with secondary antibodies. The following secondary antibodies were utilized: HRP-conjugated anti-goat (1:5000), HRP-conjugated anti-rabbit (1:1000), HRP-conjugated anti-mouse (1:1000), Alexa Fluor® 680-conjugated anti-rabbit (1:15000), and Alexa Fluor® 790-conjugated anti-mouse (1:15000). For HRP-conjugated secondary antibody-conjugated membranes, a Lumi Q ECL reagent solution kit (Share-Bio) was used for detection with a ChemiDoc™ XRS + system (BIO-RAD). The exposure was from 1 s and 1 time per second. For Alexa Flour®-conjugated secondary antibody-conjugated membranes, bound secondary antibodies were detected with an Odyssey imaging system (LI-COR Biosciences, Lincoln, NE).

## Dynabead immunoprecipitation

Primary antibodies or IgG were diluted in 200 µl ice-cold PBST (0.02% Tween 20) to the same concentration. Then, 50 µl protein G Dynabeads were precleaned and mixed with diluted antibodies or IgG and rotated at room temperature for 25 min. Primary antibodies targeting the following proteins were used: ROBO1 (N-terminal) (1:50), ROBO1 (C-terminal) (1:100), p38α (1:100), and MEK3/6 (1:50).

Cells were lysed in Cell Lysis Buffer for Western and IP (New Cell & Molecular Biotech; P70100) and Protease and Phosphatase Inhibitor Cocktail (New Cell & Molecular Biotech; P002) on ice for 10 min before centrifugation at $12,000 \times g$ for 15 min at 4 °C. Then, lysates were mixed with antibody-conjugated beads at room temperature for 30–60 min, followed by immunoblotting assessment.

### Data collection and analysis

ChemiDocTM XRS + system (BIO-RAD) and Odyssey imaging system were used for WB imaging; Flow cytometry signals were detected in BD Fortessa FACS with FACSDiva software v6.0. CT combined 3D Organ Reconstruction Bioluminescence Imaging was performed on Caliper Life Sciences; IF was observed on Leica TCS SP8 LAS X; CCK-8 assay was measured on Tecan Infinite microplate reader system. In vivo imaging data were collected by IVIS Spectrum version 4.5.3.

Flow cytometry data was analyzed using FlowJo Software version 10.4.2 software. Gene set enrichment analysis were performed by 4GB (64 bit) GSEA v4.01 Java Web Start (all platforms). In vivo imaging were assessed by IVIS Spectrum version 4.5.3.

IBM SPSS statistics 19.0 and GraphPad 7.0 software were used in this study. Data are presented as the mean ± S.E.M. Two-tailed Student's t test was used to evaluate differences between two groups. For evaluations of cell viability or tumour growth, one-way Repeated-Measure ANOVA was performed.

### Reporting summary

Further information on research design is available in the Nature Portfolio Reporting Summary linked to this article.

## Data availability

Source data are provided as a Source Data File. The authors claim that all data supporting this study are available in this article, Source Data file and Supplementary data. Source Data are provided with this paper. All data generated or analysed involved in this study are provided or cited in our article. For RNA-seq on mice liver metastasis the analysis data is available in the Sequence Read Archive (SRA), the number of which is PRJNA590588. This database contains 2 parts of analysis data: Panc02 cell line modelled liver metastasis part (Panc02_D3_ (1-3), Panc02_D6_ (1-3) and Panc02_D12_ (1-3)) and KPC1199 cell line modelled liver metastasis part (D3_ (1-3), D8_ (1-3) and D15_ (1-3)). For ROBO1 and SLIT2 expression analysis, the datasets is GSE (https://www.ncbi.nlm.nih.gov/geo/query/acc.cgi?acc=GSE71729); For EMT pathway analysis, the datasets are GEODATASET GSE15471 and https://portal.gdc.cancer.gov/projects/TCGA-PAAD. Source data are provided with this paper.

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

## Acknowledgements

This study was supported by the National Natural Science Foundation of China (82230087, 92168111), the Shanghai Municipal Education Commission—Gaofeng Clinical Medicine Grant Support (20181708), Program of Shanghai Academic/Technology Research Leader (19XD1403400), Shanghai International Science and Technology Cooperation Fund (18410721000), Excellent Academic Leader of Shanghai Municipal Health Bureau (2018BR32), Medicine and Engineering Interdisciplinary Research Fund of Shanghai Jiao Tong University (YG2021ZD08), Innovative research team of high-level local universities in Shanghai (SHSMU-ZDCX20210802), Shanghai Pilot Program for Basic Research - Shanghai Jiao Tong University (21TQ1400225) hosted by Zhigang Zhang; China Postdoctoral Science Foundation (2018M640403), National Natural Science Foundation of China (81701945) hosted by Xu Wang; National Natural Science Foundation of China (82073023, 81871923), Shanghai Municipal Education Commission—Gaofeng Clinical Medicine Grant Support (20191809) hosted by Jun Li; National Natural Science Foundation of China (81802890), Natural Science Foundation of Shanghai (18ZR1436900) hosted by Xueli Zhang; National Natural Science Foundation of China (82002485) by Qing Li; National Natural Science Foundation of China (82103357), Shanghai Sailing Program (21YF1445200), Natural Science Foundation of Shanghai (21ZR1461300) by Lipeng Hu; National Natural Science Foundation of China (8210364), the China Postdoctoral Science Foundation (2022M711386), Natural Science Foundation for Colleges and Universities in Jiangsu Province, China (21KJB320013) by Xiaoxin Zhang; National Natural Science Foundation of China (81874175) by Yongwei Sun.

## Author contributions

Z.G.Z., J.L., X.X.Z., J.R.G. and Q.L. were responsible for concept and experimental design; Y.W.S., B.N., Y.S.J., C.J.X. and M.W.Y. were responsible for clinical samples collection; Q.L., L.P.H., X.W. and S.H.J. were responsible for animal experiments; X.L.Z., Y.L.Z., P.Q.H., Y.Z. and Q.Y. performed cell molecular biology experiments, X.X.Z, X.L.Z., L.P.H., G.G.S.X. and S.H.J. were responsible for data measurement and scientific writing, H.L. and D.X.L. were responsible for paper revision.

## Competing interests

The authors declare no competing interests.
