## [Peer Review File · Nature Communications]

Coadaptation fostered by the SLIT2-ROBO1 axis facilitates liver metastasis of pancreatic ductal adenocarcinomaREVIEWER COMMENTS

Reviewer #1 (Remarks to the Author): Expert in pancreatic cancer mouse models

The data presented in this manuscript are interesting, even if the general concept is already known (shown for other cancer types). Moreover, the general topic addressed in this study (identifying new targetable mechanisms involved in pancreatic cancer dissemination to distant organs) is of particular importance as pancreatic ductal adenocarcinoma is still a lethal malignancy with around 50% of diagnosed patients presenting metastatic disease. However, some points need to be addressed before resubmission to Nat Comm.

To summarize, the authors report that the SLIT2/ROBO1 axis facilitates outgrowth of liver metastatic niches in pancreatic cancer mouse models (and confirmed it in human PDAC samples), by driving the co-evolution of hepatocytes and disseminated pancreatic cancer cells during metastatic process. The authors additionally demonstrate that ROBO1 display dependence receptor characteristics and thus, belongs to this receptor functionally related family. They finally show that ROBO1/SLIT2 interaction induces cell proliferation via a MEK3/6-p38alpha MAPK signaling pathway activation.

Even if the study is well-conducted, it needs to be revised and improved before acceptance for publication in Nat Comm.

Comments to the authors

In the present manuscript, the authors report that the SLIT2/ROBO1 axis facilitates outgrowth of liver metastatic niches in pancreatic cancer mouse models (and confirmed it in human PDAC samples), by driving the co-evolution of hepatocytes and disseminated pancreatic cancer cells during metastatic process. The authors additionally demonstrate that ROBO1 display dependence receptor characteristics and thus, belongs to this receptor functionally related family. Mechanistically, the authors also show that ROBO1/SLIT2 interaction induces cell proliferation via a MEK3/6-p38alpha MAPK signaling pathway activation. The general topic addressed in this study is of particular importance and the experiments were well-conducted.

Indeed, I very much appreciate and value their approach employing several mouse models and systems, as well as human PDAC data to support their findings. Overall, even if the general concept is already known and the involvement of SLIT2/ROBO1 axis expected (shown for other cancer types), the main results are very interesting and the manuscript would be a valuable contribution to the pancreatic cancer field, identifying a putative new targetable mechanism involved in pancreatic cancer dissemination. However, some points still need to be addressed and clarified.

Major points

1. Can the authors observe phenotypic/molecular changes in ROBO1+ metastatic cells niched at the liver metastasis site when compared to ROBO1+ tumor cells at primary tumor site? In other tumor entities, the current knowledge on SLIT2/ROBO1-mediated cell adhesion notably involves an increased stabilization/expression of E-cadherin at the membrane (E-cadherin/beta-catenin complex) and an inhibition of CDC42. Can the authors provide more molecular clues to address phenotypic/molecular changes (in cell adhesion and migration processes) in metastatic cells?
2. The authors show that ROBO1 is a dependence receptor. Its expression suggests that, to survive at the primary tumor site and possibly disseminate, cancer cells need to interact with a ROBO1 ligand and/or display alterations in this pathway. Within the primary tumor, what ROBO1 ligand is expressed and by what cell type? Is it an autocrine production and secretion of the ligand? Indeed, the authors show that PG1 and PG2 cells also weakly express SLIT2 (Extended Figure 7c). Can the authors address that particular point in human primary tumors and KPC tumors and discuss it?
3. The authors show that, in primary tumor sites, distinct ROBO1+ and ROBO1- tumor niches co-exist (Figure 5a-5b). Can they perform co-stainings showing that ROBO1- cells are indeed tumor cells and not stromal cells? Rising then the question of tumor heterogeneity, could the authors e.g. re-analyze publicly available single-cell RNA sequencing performed on human PDACs to assess heterogeneity between ROBO1+/ROBO1- tumor cell subpopulations? E.g. do the ROBO1+ tumor

cells show more mesenchymal features than ROBO1- ones?

4. The abstract is too much organized as a "catalog". It must be completely rephrased to improve its readability and informativeness and more clearly expose the main findings of the manuscript.

5. References to literature are missing in the introduction section (e.g. line 48-54) and in the results section e.g. line 157-158 and 159-160. The reviewer would recommend to carefully double-check the cited literature and the manuscript to avoid referencing to already known knowledge without an appropriate citation.

6. In the introduction section, some sentences are really unclear: e.g. sentences line 63-66 and line 66-68 should be rephrased to improve their clarity. The sentence line 68-69 has also to be developed and rephrased. I would then kindly recommend an extensive editing of English language and proofreading to substantially improve the clarity of the manuscript.

7. Several studies have already addressed the roles of dependence receptors during tumor initiation and progression and thus, a lot is already known on their roles (expression alterations, during tumor progression...). The authors should further develop, in the introduction section, the description of their already-known roles to generally improve the understanding of their study background.

8. The authors wrote "Here we filtered out AG molecular SLIT2 in our models which were overexpressed by hepatocytes through PMN to MMN and demonstrated for the first time that its classic receptor ROBO1 was a DR" (line 72). This sentence is once again unclear. Could they clarify (e.g. what models they are talking about) and develop that part?

9. The authors should further develop their final brief model cartoon (Figure 7I) to better summarize their SLIT2/ROBO1-mediated process including the mechanistic/molecular findings.

Minor points

1. To improve readability, the authors should expand each abbreviation used in the manuscript the first time it is mentioned, even if already expanded in the abstract section.

2. The authors must ensure that each abbreviation used in the figures and their corresponding legends is expanded in each figure legend.

3. Line 176, the reference to Figure 3h-3i should be corrected for Figure 3i-3k.

Reviewer #2 (Remarks to the Author): Expert in Robo/Slit pathway

This study investigates the mechanisms by which pancreatic ductal adenocarcinoma (PDAC) metastasize to the liver. The authors found the axon guidance molecules, SLIT2 and ROBO1, to be enriched in the pre-metastatic niche in both mouse cancer models and human cancer samples (primary PDAC and matched liver metastasis). They performed a set of in vivo experiments using Slit2 conditional KO mice, KPC mouse model and ROBO1-blocking antibody to assess the impact of this pathway in liver metastasis formation. Moreover, in vitro experiments in tumor cells showed a crosstalk with p38 MAPK signaling pathway being downstream of SLIT2-ROBO1. Based on these findings, the authors suggest a model in which ROBO1 works as a dependence receptor (DR) promoting cell death in the absence of the SLIT2 ligand and facilitating cell survival (metastasis outgrowth) in the presence of the ligand.

Understanding how pancreatic cancer cells exploit components of the microenvironment and spread to the liver is of high values for developing a therapeutic strategy towards metastatic PDAC. So, the question is relevant, but the manuscript in its current form is sketchy, it presents a series of observations that might be interesting but are too preliminary. Finally, the data are incomplete in many places and do not justify the conclusions made. I have listed below some specific concerns:

1) The paper claims that the hepatocytes are the source of SLIT2, which in turn would create a microenvironment favourable for ROBO1+ cancer cells to implant. However, the expression of SLIT2 in the liver is not thoroughly documented here. This study is almost entirely based on antibodies against SLIT2 and ROBO1, which are known to be not always reliable for immunohistochemistry on mouse tissues (see Pinho et al. Nat Comms 2018, Escot et al Nat Comms 2018). Different groups explored the use of antibodies against ROBO and SLIT proteins extensively and could only conclude that they do not work on mouse samples. In addition, Slit2 has been previously reported to localize in the liver in HSCs and Kupffer cells, but not in hepatocytes (see Chang et al 2015 and Coll et al 2021). Therefore, the authors should repeat their analyses using RNAscope in situ hybridisation together with cell type-specific markers to exactly define in which cells of the liver Slit2 is expressed.

2) The Slit2fl/fl; Alb-Cre in vivo model lacks important controls. Related to Point 1, the authors should show the expression of Slit2 in hepatocytes before and after cancer cells intrasplenic injection. Western Blots experiments (see Extended Data Fig.4) do not prove that Slit2 is expressed in hepatocytes, it could be instead in liver non-parenchymal cells. Western Blots needs to be corroborated by RNAscope in situ [and Immunofluorescence staining - once antibodies are validated- see above-] on the different liver samples (control, transgenic intra-splenic injected and not, rescued etc...). If Slit2 is not expressed in hepatocytes, then what is the relevance of using this model and of the reported observations?

3) Rescue experiments in Slit2fl/fl; Alb-Cre mice are done by systemic injection of lentivirus GV348 (Ubi MCS SV40) construct, whose expression is then activated by the Alb-Cre. How stable is the lentivirus once injected? This should be controlled not only by checking the levels of Slit2 (see Point 2) but also using an independent mean, for example a sequence element or epitope tag in the vector.

4) Neutralizing antibodies to ROBO1 have been shown to suppress tumor growth and metastasis by inhibiting tumor-induced angiogenesis (Wang et al 2003 and in breast cancer, prostate cancer, etc...). Can this be completely ruled out here in this model of PDAC metastasis? Moreover, the experiments validating the ROBO1 neutralizing antibody that they produced and used in this study are missing. The antibody is administered in vivo by ip injection in mice, how stable is its activity? Is there a functional read-out for the antibody activity? The effects of ROBO1 neutralizing antibody should be compared to IgG control injection experiments instead of PBS injection.

5) The manuscript text needs substantial editing for grammatical errors and use of unclear terminology. The current version is not understandable.

Reviewer #3 (Remarks to the Author): Expert in metastatic pancreatic cancer models

In this manuscript, the authors seek to determine pre- and post-metastatic environmental factors that facilitate metastatic outgrowth of pancreatic ductal adenocarcinoma (PDAC). This is important because widespread metastatic outgrowth is a poorly understood process that remains the ultimate cause of death for most patients who present with this highly lethal disease.

The authors use an intrasplenic injection experimental model to investigate changes in the hepatic microenvironment that occur during initial colonization of injected tumor cells (disseminated tumor cells: DTCs) and during the subsequent outgrowth of macroscopic metastatic tumors that develop from DTCs. The authors define these two phases as the "pre-metastatic niche" (PMN) and "macro metastatic niche" (MMN) respectively. The authors identify up-regulation of axon guidance genes in the PMN and MMN, which is consistent with previous reports that have also documented axon guidance as a core signaling pathway in PDAC. While other studies have focused on how axon guidance pathways might facilitate perineural invasion (a known property of primary PDAC), the authors investigate how axon guidance pathways might facilitate metastatic outgrowth. The authors convincingly show that hepatocytes in the "PMN/MMN" expressed the secreted ligand

SLIT2 while the PDAC cells expressed the SLIT2 receptor ROBO1. The authors then conduct several in vivo experiments to interrogate the importance of the ROBO-SLIT axis during formation of the PMN and subsequent metastatic outgrowth within the context of their injection model.

The major strength of this work is the large number of complementary in vivo experiments with confirmation of results on patient tissue samples. The data are for the most part clean and intriguing. However, there are serious concerns about heavy reliance on over-expression models that could significantly skew experimental outcomes toward biased results (that fit with the author's hypothesis/model). The text is also very difficult to read and review, and the authors definition of a PMN within the context of their experimental model is questionable.

General Comments

1. Although the data figures are presented well, the manuscript text is very poorly written in broken English. The authors should request an English proficient colleague to proof and revise the language of the manuscript for clarity.

2. In this work a pre-metastatic niche (PMN) is defined by response of target organ to injected tumor cells (as a model of DTCs). This seems confusing as a pre-metastatic niche is generally thought to develop in response to non-cellular factors (e.g. exosomes, cytokines) that are secreted from the primary tumor into the bloodstream that then remodel the target organ into a microenvironment that is fertile for secondary outgrowth of DTCs.

3. Tumor-hepatocyte "co-evolution" and similar jargon is easily confused with other more common terms in the literature (especially genetic subclonal evolution of tumor cells). Tumor-hepatocyte "interaction" or "cross-talk" seem more appropriate and less confusing.

4. Although the reviewer appreciates the large number of in vivo experiments, it is nevertheless unclear why most of these experiments relied so heavily on over-expression of exogenous Robo1 ("Robo1-FL" cells). Strong rationale for this intervention should be provided, especially since the parental cells ("CTRL") appear to form metastases on par with the Robo1-FL cells (Figure 6b for example). Especially concerning is the possibility that overexpressing exogenous Robo1 could render these cells artifactually dependent on SLIT2, thereby heavily biasing the experimental results toward the proposed Robo1-Slit2 cross talk model. Such a dependence is implied by in vitro data shown in Figure 6 and Extended Data Figure 10.

Technical Comments

Figures 1-2: IHC markers used by the authors to define the "metastatic niche" consist of myeloid/macrophage markers (CD168, CD68, CD11b). The figures clearly show that the IHC signals are localized to the reticuloendothelial lining of the hepatic sinusoids, presumably labeling activated Kupffer cells that reside there. Negative control experiments (such as injection of non-PDAC cells) should be conducted to ensure that the PMN changes are specific to the metastatic process and not simply a generic response to the presence of foreign cells/material.

Figure 4: The Robo1-FL cells should be characterized, since much of the experimental data rely on these cells. At the very least, expression data (relative to parental cells) should be shown, and other pertinent assays conducted to characterize the properties of these cells more fully in relation to the parental control cells from which they were derived (in vitro proliferation rates vs parental cells for example). Some of these data are loosely shown at various (seemingly random) panels in later Figures, but these should be consolidated to earlier in the manuscript and incorporated into the rationale for over-expressing Robo1.

Figure 4: Although the reviewer appreciates the in vivo SLIT2 loss-of-function with rescue, it conversely seems that a similar loss-of-function approach knocking out endogenous Robo1 in the pancreatic cancer cells would be a superior experimental approach to overexpressing exogenous Robo1. Such experiments would also provide a rigorous way to ensure that the systemically administered anti-Robo1 antibody results are not indirect or off-target effects.

Figure 5a: H&E stains corresponding to the IHC stains should be shown to highlight location of the tumor glands with respect to the Robo1 IHC signals (especially for the primary tumors on the right

panels, case 5-8).

Figure 5c-e: These experiments again rely on over-expression of exogenous Robo1 (the "Robo1-FL" cells), which could introduce artificial biases into the results. Rigor would be enhanced if the parental ("CTRL") Panc02/Kpc1199 cells were used instead, to see if cells expressing high Robo1 are naturally (rather than artificially) selected during the serial metastasis passaging. Proper in vitro control experiments are also missing to ensure that Robo-FL cells do not simply proliferate at faster basal rates (and Δ Robo1 cells at slower rates) than the parental CTRL cells.

Figure 6f-g, Extended Data Figure 10: Why is the initial viability of cells only ~40% (D0)? Why is FBS then removed from the media once the cell viability begins to rise (D2)? Why do all the cells then begin dying at approximately the same rate (D2-D6)? There is some reference to a "cell stress exertion" in the methods section, but this experiment is described in the manuscript text (line 292) as showing slower growth rates (rather than viability, which is plotted, or "cell stress exertion" which is mentioned in the methods). It is not clear what is being measured (what is the readout?) or what they authors are trying to show (viability or proliferation rates?). This is a very confusing experiment.

Point-by-point response:

Reviewer #1

Comment: 1. Can the authors observe phenotypic/molecular changes in ROBO1+ metastatic cells niched at the liver metastasis site when compared to ROBO1+ tumor cells at primary tumor site? In other tumor entities, the current knowledge on SLIT2/ROBO1-mediated cell adhesion notably involves an increased stabilization/expression of E-cadherin at the membrane (E-cadherin/beta-catenin complex) and an inhibition of CDC42. Can the authors provide more molecular clues to address phenotypic/molecular changes (in cell adhesion and migration processes) in metastatic cells?

Response: Thank you for your suggestions. We have added IF experiments to examine EMT markers E-cadherin, N-cadherin and CDC42 expression at primary PDAC niches as well as liver metastasis. In summary, the activation of SLIT2-ROBO1 axis could increase expression of E-cadherin and inhibit CDC42 (Response Figure 1). This is in accordance with our findings: SLIT2-ROBO1 activation could promote tumor cells' implantation. Data have been added in Extended data Figure 4c

Response Figure 1

Comment: 2. The authors show that ROBO1 is a dependence receptor. Its expression suggests that, to survive at the primary tumor site and possibly disseminate, cancer cells need to interact with a ROBO1 ligand and/or display alterations in this pathway. Within the primary tumor, what ROBO1 ligand is expressed and by what cell type? Is it an autocrine production and secretion of the ligand? Indeed, the authors show that PG1 and PG2 cells also weakly express SLIT2 (Extended Figure 7c). Can the authors address that particular point in human primary tumors and KPC tumors and discuss it?

Response: Thank you for rising this concern. To solve this problem, we have performed IF staining on primary tumor samples of both PDAC patients and KPC mouse models. We have discovered that in primary site, SLIT2 is mainly derived from cancer associated fibroblasts (CAFs) which are usually abundant in primary PDAC tumors (Response Figure 2). Data have been added in Extended data Figure 4a-4b.

Response Figure 2

Comment:3. The authors show that, in primary tumor sites, distinct ROBO1+ and ROBO1- tumor niches co-exist (Figure 5a-5b). Can they perform co-stainings showing that ROBO1- cells are indeed tumor cells and not stromal cells? Rising then the question of tumor heterogeneity, could the authors e.g. re-analyze publicly available single-cell RNA sequencing performed on human PDACs to assess heterogeneity between ROBO1+/ROBO1- tumor cell subpopulations? E.g. do the ROBO1+ tumor cells show more mesenchymal features than ROBO1- ones?

Response: Thank you for advising. We used serial sections to show ROBO1⁺ niches (ROBO1 IHC-P and matched H&E staining) (Response Figure 3). Now it is easier to distinguish tumour niches and non-tumour area via observing H&E staining sections, which also made it possible to recognize ROBO1+ and ROBO1- tumor niches combining ROBO1 stained IHC-P. Data have been added in Figure 5a.

For database analyzing, we feel sorry that we have done our best to search for a long time, only finding GSE156405 which contained liver metastatic lesion. Due to the limitation of single cell RNA sequencing, we could not gain enough data to proceed. Instead, we have utilized GSE15471 (17 cases Primary PDAC and 17cases matched adjacent normal pancreas) and TCGA PAAD database (179 cases primary PDAC). We performed analysis according to ROBO1 expression. GSE15471 (17 ROBO1-high expression vs. 17 ROBO1-Low expression) and TCGA (60 ROBO1-high expression vs. 60 ROBO1-Low expression). We discovered EMT pathway enrichment in both databases (Response Figure 4). Data have been added in Extended data Figure 4d-4e.

Response Figure 3

Response Figure 4

Comment:4. The abstract is too much organized as a “catalog”. It must be completely rephrased to improve its readability and informativeness and more clearly expose the main findings of the manuscript.

Response: Thank you for your kind suggestion. We now have rewrite the abstract as you have suggested.

Comment:5. References to literature are missing in the introduction section (e.g. line 48-54) and in the results section e.g. line 157-158 and 159-160. The reviewer would recommend to carefully double-check the cited literature and the manuscript to avoid referencing to already known knowledge without an appropriate citation.

Response: Thank you for your kind suggestion. We feel sorry for these mistakes. We now have added proper reference on them.

Comment:6. In the introduction section, some sentences are really unclear: e.g. sentences line 63-66 and line 66-68 should be rephrased to improve their clarity. The sentence line 68-69 has also to be developed and rephrased. I would then kindly recommend an extensive editing of English language and proofreading to substantially improve the clarity of the manuscript.

Response: Thank you for your kind suggestion. We feel sorry for these mistakes. We have rewritten the mentioned parts and we have also improved the English-writing of whole paper.

Comment:7. Several studies have already addressed the roles of dependence receptors during tumor initiation and progression and thus, a lot is already known on their roles (expression alterations, during tumor progression...). The authors should further develop, in the introduction section, the description of their already-known roles to generally improve the understanding of their study background.

Response: Thank you for rising concern. We now have added findings of DRs in these proper sites for readers to realize their functions.

Comment:8. The authors wrote “Here we filtered out AG molecular SLIT2 in our models which were overexpressed by hepatocytes through PMN to MMN and demonstrated for the first time that its classic receptor ROBO1 was a DR” (line 72). This sentence is once again unclear. Could they clarify (e.g. what models they are talking about) and develop that part?

Response: Thank you for rising concern. We now rewrite this sentence as “Here, we filtered out the AG molecule SLIT2 in our models, which was overexpressed by hepatocytes through progression of the PMN to the MMN and demonstrated for the first time that its classic receptor ROBO1 is a DR.” which might be clearer.

Comment:9. The authors should further develop their final brief model cartoon (Figure 71) to better summarize their SLIT2/ROBO1-mediated process including the mechanistic/molecular findings.

Response: Thank you for your suggestions. We now have added brief cartoon to explain SLIT2-ROBO1 axis (Response Figure 5). This figure has been added in Figure 8b

Response Figure 5

Comment:10. To improve readability, the authors should expand each abbreviation used in the manuscript the first time it is mentioned, even if already expanded in the abstract section.

Response: Thank you for your suggestions. We have expanded abbreviation used in manuscript when they were mentioned for the first time.

Comment:11 The authors must ensure that each abbreviation used in the figures and their corresponding legends is expanded in each figure legend.

Response: Thank you for your suggestions. We have expanded abbreviation used in the figures in their corresponding legends.

Comment:12. Line 176, the reference to Figure 3h-3i should be corrected for Figure 3i-3k.

Response: Thank you for your suggestions. We have corrected this mistake.

Reviewer #2

Comment:1) The paper claims that the hepatocytes are the source of SLIT2, which in turn would create a microenvironment favourable for ROBO1+ cancer cells to implant. However, the expression of SLIT2 in the liver is not thoroughly documented here. This study is almost entirely based on antibodies against SLIT2 and ROBO1, which are known to be not always reliable for immunohistochemistry on mouse tissues (see Pinho et al. Nat Comms 2018, Escot et al Nat Comms 2018). Different groups explored the use of antibodies against ROBO and SLIT proteins extensively and could only conclude that they do not work on mouse samples.

In addition, Slit2 has been previously reported to localize in the liver in HSCs and Kupffer cells, but not in hepatocytes (see Chang et al 2015 and Coll et al 2021).

Therefore, the authors should repeat their analyses using RNAscope in situ hybridisation together with cell type-specific markers to exactly define in which cells of the liver Slit2 is expressed.

Response: Thank you for your suggestions. We have fully realized that the expression and distribution of SLIT2 should be examined by RNAscope. Here we performed RNA scope on liver metastasis samples of PDAC patients, we co-detected *SLIT2* mRNA with *ALB* mRNA (hepatocyte marker), *KRT19* mRNA (tumor cell marker) and *ATCA1* mRNA (cancer associated fibroblast marker). Results have demonstrated that in liver metastatic niches of PDAC, the main source of SLIT2 is adjacent hepatocytes which co-adapted with tumour cells but not CK19 expressed tumor cells or α -SMA expressed CAFs. The results have confirmed our hypothesis (Response Figure 6).

And these data have been added to Figure 3f

Response Figure 6

Comment:2) The *Slit2*^{fl/fl}; *Alb-Cre* in vivo model lacks important controls. Related to Point 1, the authors should show the expression of *Slit2* in hepatocytes before and after cancer cells intrasplenic injection. Western Blots experiments (see Extended Data Fig.4) do not prove that *Slit2* is expressed in hepatocytes, it could be instead in liver non-parenchymal cells. Western Blots needs to be corroborated by RNAscope in situ [and Immunofluorescence staining - once antibodies are validated- see above-] on the different liver samples (control, transgenic intra-splenic injected and not, rescued etc...).

If *Slit2* is not expressed in hepatocytes, then what is the relevance of using this model and of the reported observations?

Response: Thank you for your suggestions. We have also recognized that only WB validation could not be convincing. We now added RNAscope results to detected *Alb* and *Slit2* mRNA in

CTRL/PMN/MMN formed livers of *Kpc1199^{Robo-FL}* injected mouse model. Data have illustrated that in *Kpc1199* injection mouse model, SLIT2 is derived from hepatocytes around the metastatic niches. We also used RNAscope to examine *Alb* and *Slit2* mRNA in *Kpc1199^{Robo-FL}* and *Panc02^{Robo-FL}* injected SLIT2 CKO and SLIT2 CKO-RE mouse models (Response Figure 7). Results have demonstrated that the injection of tumor cell could stimulate the SLIT2 expression in CTRL and CKO-RE groups (Response Figure 7). And data have been added in Figure 3f, Extended Data Figure 6d and Extended Data Figure 7b.

Response Figure 7

Comment:3) Rescue experiments in Slit2^{fl/fl}; Alb-Cre mice are done by systemic injection of lentivirus GV348 (Ubi MCS SV40) construct, whose expression is then activated by the Alb-Cre. How stable is the lentivirus once injected? This should be controlled not only by checking the levels of Slit2 (see Point 2) but also using an independent mean, for example a sequence element or epitope tag in the vector.

Response: Thank you for your advice. We have noticed that it is of importance to confirm the stability of this expressing system. To consolidate, we constructed a same lentivirus system expressing SLIT2-GFP fusion protein. We selected day0 (D0), day (D14), day21 (D21) and day28 (D28) these 4 time points to check it out. We have found that either SLIT2 or GFP could be detected in livers of SLIT2-CKO mice from D0 to D28 (Response Figure 8). These data could also be available in Extended Data Figure 6c.

Response Figure 8

Comment: 4.1 Neutralizing antibodies to ROBO1 have been shown to suppress tumor growth and metastasis by inhibiting tumor-induced angiogenesis (Wang et al 2003 and in breast cancer, prostate cancer, etc...). Can this be completely ruled out here in this model of PDAC metastasis?

Response: Thank you for rising this concern. We have fully known that SLIT2-ROBO1 axis could positively facilitated angiogenesis as you have mentioned. While in this intrasplenic injection model, the formation of PMN and MMN is much faster than usual model (such as KPC mouse model), it is more possible that ROBO1⁺ tumor cells gained the implantation and outgrowth

advantage via enhancing the cell functions of their own rather than generate angiogenesis. Especially, PMN (in which stage DTCs have just arrived at metastatic sites) could formed earlier than angiogenesis, while the antibody against ROBO1 could attenuate tumor growth from PMN stage. Besides, antibody targeting ROBO1 could also hamper growth of tumor cells *in vitro*, which further consolidated that this function might be independent of anti-angiogenesis.

Comment: 4.2 Moreover, the experiments validating the ROBO1 neutralizing antibody that they produced and used in this study are missing. The antibody is administered *in vivo* by ip injection in mice, how stable is its activity? Is there a functional read-out for the antibody activity?

Response: Thank you for your suggestions. We attached importance to this advice. To consolidate this antibody, we used antibodies at different concentrations (80µg, 200µg and 500µg) to treat our mouse model. We have gained that 500µg is the best concentration. We have also compared the difference in interval time of administration, finding that injection every 3 days could significantly extended the median survival time of mouse model. Additionally, we have performed PANC1 injection mouse model followed by antibody treatment and gained the same results (Response Figure 9). And these data have been added to Extended Data Figure 8.

Response Figure 9

Comment: 4.3 The effects of ROBO1 neutralizing antibody should be compared to IgG control injection experiments instead of PBS injection.

Response: Thank you for rising this concern. We do apologize that it was our fault to use PBS as control. We have deleted all figures in which PBS was performed as control group to antibody. We then re-designed and repeated all these experiments via using IgG as control group and we have put new figures on the paper. The related figures are: Figure 4e-Figure 4i, Figure 6g, Extended Data Figure 9a-9g, Extended Data Figure 13d.

Comment: 5 The manuscript text needs substantial editing for grammatical errors and use of unclear terminology. The current version is not understandable.

Response: Thank you for rising this concern. We have rewritten our manuscript to improve readability.

Reviewer #3

Comment:1. Although the data figures are presented well, the manuscript text is very poorly written in broken English. The authors should request an English proficient colleague to proof and revise the language of the manuscript for clarity.

Response: Thank you for your suggestions. We have rewritten this article to improve our English writing.

Comment:2. In this work a pre-metastatic niche (PMN) is defined by response of target organ to injected tumor cells (as a model of DTCs). This seems confusing as a pre-metastatic niche is generally thought to develop in response to non-cellular factors (e.g. exosomes, cytokines) that are secreted from the primary tumor into the bloodstream that then remodel the target organ into a microenvironment that is fertile for secondary outgrowth of DTCs.

Response: Thank you for rising concerns. As you have mentioned, PMN is known as a complex tumour microenvironment, the formation of which relies on TME cells, secreted proteins, and tumour cells. In some study (Héctor Peinado et al., 2017), researchers have defined PMN as several stage as “early PMN” and “evolving stage”. The mechanisms involved are different, and non-cellular factors and cellular factors are equally important for PMN formation, and the latter might play important roles in evolving stage of PMN. There are also reports disseminated tumor cell arrive at metastatic site much earlier while stay in a dormant state to wait for microenvironment remodelling , could also be defined as PMN. This mechanism could also be

both non-cellular and cellular (Arnaud Pommier et al., 2018). We used the intrasplenic mouse model for PMN and MMN detection because the time point (PMN stage and MMN stage) of this model is relatively stable and repeatable. We have also attempted to perform orthotopic injection of tumor cells on pancreas of mice and induced spontaneous liver metastasis. While it is hard to control or observe the time point of PMN and MMN in liver as well as in KPC mice. Though this model has some limitations (such as a relative rapid early PMN stage), while there are still some merits in it. We have done our best to consolidate the results, and see also in our response in your Comment 5.

Comment:3. Tumor-hepatocyte “co-evolution” and similar jargon is easily confused with other more common terms in the literature (especially genetic subclonal evolution of tumor cells). Tumor-hepatocyte “interaction” or “cross-talk” seem more appropriate and less confusing.

Response: Thank you for your suggestions. We now used “Co-adaption” which is more proper to describe the reciprocal interaction of tumor cells and hepatocytes mediated by SLIT2 and ROBO1.

Comment:4. Although the reviewer appreciates the large number of in vivo experiments, it is nevertheless unclear why most of these experiments relied so heavily on over-expression of exogenous Robo1 (“Robo1-FL” cells). Strong rationale for this intervention should be provided, especially since the parental cells (“CTRL”) appear to form metastases on par with the Robo1-FL cells (Figure 6b for example). Especially concerning is the possibility that overexpressing exogenous Robo1 could render these cells artifactually dependent on SLIT2, thereby heavily biasing the experimental results toward the proposed Robo1-Slit2 cross talk model. Such a dependence is implied by in vitro data shown in Figure 6 and Extended Data Figure 10.

Response: Thank you for your suggestions. We have also realized that artifactually dependence of SLIT2 could be induced if exogenous ROBO1 was always performed. We now have designed more experiments using endogenous ROBO1 expressed cell PANC1 to consolidate our data. The results could be available in the latter response to comments.

Technical Comments

Comment:5 Figures 1-2: IHC markers used by the authors to define the “metastatic niche” consist of myeloid/macrophage markers (CD168, CD68, CD11b). The figures clearly show that the IHC signals are localized to the reticuloendothelial lining of the hepatic sinusoids, presumably labeling

activated Kupffer cells that reside there. Negative control experiments (such as injection of non-PDAC cells) should be conducted to ensure that the PMN changes are specific to the metastatic process and not simply a generic response to the presence of foreign cells/material.

Response: Thank you for your suggestions. To solve this problem, we have added another experiment to make the detection of PMN more solid. Here we used a murine PDAC cell line LTPA as a negative control, which could not form liver metastasis in liver after intrasplenic injection. We have also used PBS for injection as Sham group. Results have shown that CD68 positive macrophages could be observed at D8 in all group, while in Kpc1199 group, the amount of which significantly increased, indicating the CD68 positive macrophage in Kpc1199 group were not all kupffer cells. We also detected M2 TAMs marker CD163, results have displayed that CD163 positive TAMs aggregated in KPC199 induced PMN, but not in other two groups. And LOX expression could only be observed in Kpc1199 group (Response Figure 10). These data have also been added to Figure 1i.

Response Figure 10

Comment:6 Figure 4: The Robo1-FL cells should be characterized, since much of the experimental data rely on these cells. At the very least, expression data (relative to parental cells) should be shown, and other pertinent assays conducted to characterize the properties of these cells more fully in relation to the parental control cells from which they were derived (in vitro proliferation rates vs parental cells for example). Some of these data are loosely shown at various (seemingly random) panels in later Figures, but these should be consolidated to earlier in the

manuscript and incorporated into the rationale for over-expressing Robo1.

Response: Thank you for your suggestions. We have realized that we should put these data earlier in article for readability. Now we reorganized our figures to characterize ROBO1 molecular in one figure in Extended Data Figure 5, including the structure, expression, and cell function of ROBO1 expressed tumor cells.

Comment:7 Although the reviewer appreciates the in vivo SLIT2 loss-of-function with rescue, it conversely seems that a similar loss-of-function approach knocking out endogenous Robo1 in the pancreatic cancer cells would be a superior experimental approach to overexpressing exogenous Robo1. Such experiments would also provide a rigorous way to ensure that the systemically administered anti-Robo1 antibody results are not indirect or off-target effects.

Response: Thank you for your suggestions. We now used RNAi on PANC1 cell before modelled mice to evaluate the function of SLIT2-ROBO1 axis as you have suggested. We also used PANC1 modelled mice for neutralizing antibody experiment according to your advice (Response Figure 11). These data have been add in Extended Data Figure 5i-5k, Extended Data Figure 8d-8f.

Response Figure 11

Comment:8 Figure 5a: H&E stains corresponding to the IHC stains should be shown to highlight location of the tumor glands with respect to the Robo1 IHC signals (especially for the primary tumors on the right panels, case 5-8).

Response: Thank you for your suggestions. We now have remade this figure, using serial sections for IHC-P and matched H&E staining (Response Figure 12). These data have been added to Figure 5a.

Response Figure 12

Comment:9 Figure 5c-e: These experiments again rely on over-expression of exogenous Robo1 (the “Robo1-FL” cells), which could introduce artificial biases into the results. Rigor would be enhanced if the parental (“CTRL”) Panc02/Kpc1199 cells were used instead, to see if cells expressing high Robo1 are naturally (rather than artificially) selected during the serial metastasis passaging. Proper in vitro control experiments are also missing to ensure that Robo-FL cells do not simply proliferate at faster basal rates (and Δ Robo1 cells at slower rates) than the parental CTRL cells.

Response: Thank you for rising concerns. We know repeat this experiment using PANC1^{shCTRL} and PANC1^{shROBO1} for further validation. We also designed a in vitro experiment as you have mentioned. We mixed Kpc1199^{Robo1-FL} and Kpc1199^{Ctrl} or Panc02^{Robo1-FL} and Panc02^{Ctrl} together (1:1) and cultured them with or without SLIT2 administration, results have confirmed our conclusions (Response Figure 13). And these data have been added in Extended Data Figure 10b-10c, 10f.

Response Figure 13

Commnet:10 Figure 6f-g, Extended Data Figure 10: Why is the initial viability of cells only ~40% (D0)? Why is FBS then removed from the media once the cell viability begins to rise (D2)? Why do all the cells then begin dying at approximately the same rate (D2-D6)? There is some reference to a “cell stress exertion” in the methods section, but this experiment is described in the manuscript text (line 292) as showing slower growth rates (rather than viability, which is plotted, or “cell stress exertion” which is mentioned in the methods). It is not clear what is being measured (what is the readout?) or what they authors are trying to show (viability or proliferation rates?). This is a very confusing experiment.

Response: Thank you for rising concerns. The Y-axis is values of absorbance (OD) at 450nm but not survival rate. We now have noted this in figure legend. Considering that FBS would provide additional stimulation for cell proliferation, we removed it aiming to observe the effect of SLIT2 exerted on cells better. While cells would not grow in absence of FBS at D0, we allowed their growth with FBS at first 2 days. For DTCs, the alien environment always lack abundant nutrient,

we designed this experiment not only to evaluate the proliferation promotion of SLIT2-ROBO1, but also to observe whether this axis could enhance the survival abilities.

REVIEWERS' COMMENTS

Reviewer #1 (Remarks to the Author):

The authors have satisfactorily responded to my comments and questions, and made the necessary changes to the manuscript and figures.

Reviewer #3 (Remarks to the Author):

The authors have addressed my concerns. The manuscript is improved. Substantial editorial revisions to the text may be needed to improve the readability.

REVIEWERS' COMMENTS

Reviewer #1 (Remarks to the Author):

Comment 1: The authors have satisfactorily responded to my comments and questions, and made the necessary changes to the manuscript and figures.

Response: Thank you for your kind suggestions.

Reviewer #3 (Remarks to the Author):

Comment 1: The authors have addressed my concerns. The manuscript is improved. Substantial editorial revisions to the text may be needed to improve the readability.

Response: Thank you for rising concern. We have improved the readability according to instructions.